# Learning to Reconfigure: Configuration-Control Co-optimization of Reconfigurable Robots for Heterogeneous Locomotion

**Xiaoyu Xiong** [1 2]  **Kehan Liu** [2]  **Huiyi Yan** [3]  **Shengjie Wang** [2]  **Yang Gao** [2 1]  **Tao Du** [2 1]

## Abstract

Traditional robot co-design approaches typically converge to *one* configuration, which do not explore the flexibility from reconfiguration on heterogeneous environments. On the other hand, existing designs for reconfigurable robots require human-designed configurations. We present Learning to Reconfigure, a holistic pipeline for configuration-control co-optimization of reconfigurable robots in heterogeneous locomotion tasks consisting of several sub-tasks. Our pipeline proposes low-level specialized primitives with a high-level scheduler. To jointly optimize configuration design and control, our primitives employ a multi-tail architecture that disentangles these distinct objectives. Building on this, the scheduler learns to dynamically switch configurations based on global task progress. We evaluate our pipeline on locomotion tasks across walking, flying, and swimming, and compare with the state-of-the-art baselines, including single-robot control and multi-morphology co-optimization algorithms. Quantitative results based on traversal progress show that our pipeline outperforms single-robot baselines by 5.95x average progress. Compared with the reconfiguration-free design given by the co-design algorithms, our robots also exhibit 9.81x progress on average. These results highlight the critical role of configuration adaptation in achieving versatile robotic autonomy in complex worlds.

## 1. Introduction

Reconfigurable robots are characterized by their ability to dynamically alter their structure among several configurations, enabling them to adapt to heterogeneous sub-tasks within complex missions. This plasticity allows them to traverse diverse obstacles insurmountable for fixed-configuration systems, with demonstrated potential in search tasks within confined spaces (Yim et al., 2000), rescue operations across unstructured terrain (Salemi et al., 2006; Liu et al., 2023), and amphibious operations bridging multiple physical environments (Ijspeert et al., 2007). Formally, a configuration is defined as a kinematic state where a subset of joints are locked at specific angles to reduce control dimensionality. Therefore, this paper focuses on automatically co-optimizing a library of available configurations and a unified policy for switching among them, given a long-horizon locomotion task consisting of heterogeneous sub-tasks across land, water, and air and a robot embodiment. Traditional approaches in reconfigurable robotics (Kalantari & Spenko, 2013; Zhao et al., 2018) typically rely on hardware-centric modular designs with human-engineered configurations. On the other hand, recent developments in robot co-design (Yuan et al., 2021; Hu et al., 2023; Lu et al., 2025) automate the joint optimization of morphology and control, leveraging Reinforcement Learning (RL) or differentiable simulation to discover performant designs. However, these methods predominantly converge to a *single* static configuration which prematurely adapts to only part of the whole task. To the best of our knowledge, the automated co-optimization of *dynamic* reconfiguration strategies remains an unexplored frontier in the literature. This problem poses a unique challenge compared to static co-design: it requires not only optimizing configuration and control but also discovering the optimal scheduler for reconfiguration, which vastly expands the search space and complexity.

As the first attempt to tackle this challenge, we develop a holistic pipeline, *Learning to Reconfigure*, for automating the discovery of configuration design and control of reconfigurable robots. Our pipeline begins with a low-level exploration stage, where we introduce several parallel *specialized primitives* consisting of a multi-tail architecture to disentangle the optimization of configuration design and control. By training these primitives across different sub-tasks, we generate a diverse library of physically feasible configurations. Next, we freeze these primitives and train a high-level *scheduler*, which observes the global task progress

[1] Shanghai Qi Zhi Institute, Shanghai, China [2] Tsinghua University, Beijing, China [3] Xi'an Jiaotong University, Shaanxi, China. Correspondence to: Tao Du <taodu@tsinghua.edu.cn>.

*Proceedings of the 43$^{rd}$ International Conference on Machine Learning*, Seoul, South Korea. PMLR 306, 2026. Copyright 2026 by the author(s).

and learns to strategically switch the primitives, driving the robot to adapt to evolving environmental constraints. To facilitate this learning process, we present a multi-physics simulation environment based on MuJoCo supporting terrestrial and simplified fluid dynamics, enabling the unified training of terrestrial, aerial, and aquatic locomotion. These environments with diverse physical properties create a rigorous benchmark to demonstrate the necessity of dynamic reconfiguration over static designs.

We design diverse locomotion tasks spanning walking, flying, and swimming with several robot embodiments to evaluate our pipeline. We perform benchmark experiments comparing our method with two categories of baselines: state-of-the-art single-robot control algorithms (manually designed configurations, PPO (Schulman et al., 2017) and TD-MPC2 (Hansen et al., 2023)) and non-reconfigurable co-design algorithms (BodyGen (Lu et al., 2025) and GLSO (Hu et al., 2023)). Quantitative evaluations based on traversal progress in fixed time show that our method achieves **5.95x** traversal distance over single-robot baselines. Meanwhile, compared with the non-reconfigurable robots output by traditional co-design algorithms, our robots also achieve **9.81x** performance on average. Specifically, reconfiguration-free control baselines struggle on these heterogeneous tasks due to the complexity in physical properties (e.g. the incompatibility between the traction required for walking and the aerodynamics needed for flying). Similarly, non-reconfigurable robots generated by co-design algorithms fail to surmount this obstacle, converging to a single morphology with a compromise design that fails when the terrain shifts. The substantial performance gap highlights the unique value of reconfigurable robots, demonstrating that incorporating configuration design is a fundamental prerequisite for robotic versatility in complex worlds.

In summary, this work makes the following contributions:

*First*, we develop a novel, hierarchical co-optimization framework, *Learning to Reconfigure*, for automating the generation of specialized configurations and control policies for reconfigurable robots in heterogeneous locomotion tasks.

*Second*, we present a multi-tail policy architecture that jointly optimizes configuration design and control, integrated with a high-level scheduler to solve the long-horizon configuration switching problem in multi-physics tasks.

*Third*, we demonstrate the effectiveness of our pipeline quantitatively against strong state-of-the-art baselines, showcasing superior performance in heterogeneous locomotion tasks (e.g., land-air-water traversal) where dynamic adaptation is a prerequisite for success.

## 2. Related Works

**Reconfigurable robots** Reconfigurable robots change between configurations to adapt to different sub-tasks in a complex environment. We focus on locomotion tasks, a prime example where morphological adaptation allows agents to overcome environmental changes. Related research can be categorized into hardware design and control algorithms. Seminal hardware platforms like M-TRAN (Murata et al., 2003), SuperBot (Salemi et al., 2006), and SMORES-EP (Liu et al., 2023) have demonstrated versatile locomotion capabilities. In parallel, the control part follows a hierarchical structure. For the lower-level execution, approaches range from Central Pattern Generators (CPGs) (Ijspeert, 2008; Sproewitz et al., 2008) and hormone-based control(Shen et al., 2002) to recent Graph Neural Networks (GNNs) (Wang et al., 2018; Huang et al., 2020) and Transformers (Gupta et al., 2022; Kurin et al., 2020). For the higher-level reconfiguration planning, methodologies include centralized graph search (Rus & Vona, 1999), rule-based heuristics (Yim et al., 2000), distributed reconfiguration algorithms (Butler & Rus, 2003; Rubenstein et al., 2014), and hierarchical reinforcement learning (RL) methods (Pathak et al., 2019; Whitman et al., 2023). In this work, we leverage these established hardware designs and learning-based control backbones. However, a limitation in prior works (Pathak et al., 2019; Kyaw et al., 2022) is their reliance on human-designed target configurations. We distinguish our approach by automating the discovery of configurations instead of selecting from a fixed repertoire.

**Robot co-design** Robot co-design jointly optimizes robot morphologies and controllers. Regarding morphology, approaches have evolved from assembling basic primitives (Sims, 1994) and using recursive graph grammars to ensure structural plausibility (Zhao et al., 2020), to utilizing GNNs for feature encoding (Wang et al., 2019). In parallel, control strategies have progressed from optimizing gait parameters (Ijspeert, 2008) to training morphology-aware policies using deep RL (Gupta et al., 2021). For the core joint optimization algorithms, early works utilized Evolutionary Algorithms (EAs) to evolve virtual creatures (Sims, 1994; Bhatia et al., 2021); subsequent studies introduced RL to optimize designs via gradient-based methods (Yuan et al., 2021; Lu et al., 2025); and recently, diffusion models (Wang et al., 2023) have been applied. Despite these advancements, they predominantly focus on a *single configuration* optimized for a specific task distribution. Extending these methods to reconfigurable robots is non-trivial due to the dual challenge of discovering a library of functional configurations and learning a high-level policy to switch between them. Our framework addresses this by decoupling the primitive discovery from the switching policy learning.

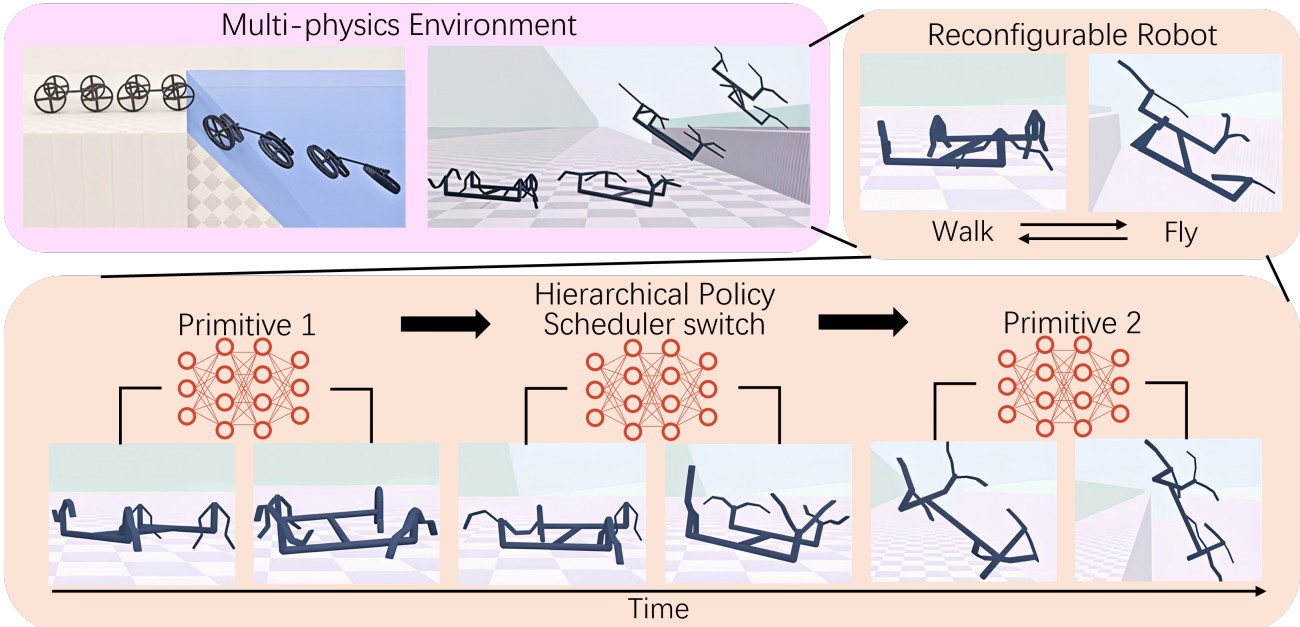

*Figure 1. Pipeline overview* (Sec. 3- Sec. 5). We propose a multi-physics environment (left-top) which supports terrestrial, aerial, and aquatic dynamics (Sec. 3) to demonstrate the performance of our co-optimization pipeline for reconfigurable robots (right-top). The pipeline utilizes a hierarchical structure (bottom) which consists of lower-level primitives for each configuration (Sec. 4) and a higher-level scheduler to integrate these configurations (Sec. 5)

**Multi-physics locomotion** Robot locomotion across diverse physical media has been widely investigated to expand the operational range of autonomous robots. Prior research has developed specialized systems for distinct environments, ranging from terrestrial walking (Hutter et al., 2016; Sproewitz et al., 2008), aerial flight (Mellinger et al., 2013; Loquercio et al., 2019), to aquatic swimming (Ijspeert et al., 2007; Katzschmann et al., 2018). Traditionally, crossing among these physical media for one robot is challenging due to their distinct mechanical features, while our pipeline decouples these different domains with several independent configurations. We validate our approach across these diverse physical fields in simulation, capable of automatically discovering specialized configurations and control policies for terrestrial, aerial, and aquatic environments.

## 3. Simulation Environment

To enable our robot to perform diverse tasks across terrestrial, aerial and aquatic domains, we propose a multi-physics simulation environment based on MuJoCo (Todorov et al., 2012). We inherit the articulated robot simulator from MuJoCo. More concretely, we build the robot with capsule and mesh links, connected with hinge joints and controlled by torques added to the joints. Besides, we build diverse terrains including tunnel, mountain, stairs, etc., reusing the collision detection from MuJoCo. We implement a simplified fluid dynamics model to simulate the interaction

between the robot and fluid media (air or water). Specifically, we model drag force $F_d$ and lift force $F_l$ acting on each link as follows:

$$F_d = -\frac{1}{2}\rho C_d A|v|v \quad F_l = \frac{1}{2}\rho C_l A|v|^2 \hat{n}_l \quad (1)$$

where $\rho$ denotes the fluid density, $v$ is the relative velocity, $A$ is the projected area, and $\hat{n}_l$ is the lift direction. Besides, we also implement a buoyancy force $F_b$ by $F_b = \rho g V_{sub}\hat{z}$, where $g$ is gravity and $V_{sub}$ is the submerged volume. Crucially, we actively detect whether each link of the robot is submerged underwater when calculating these two interactions, which helps us process the water-air interface behaviors. We combine this fluid system with the diverse terrains to form heterogeneous locomotion tasks. The full gallery of these tasks is visualized in App. B.1 Fig. 6.

## 4. Single-Configuration Primitive

Training a unified policy across heterogeneous terrains often leads to optimization bias, where the agent overfits to one sub-task (e.g., terrestrial walking) while underperforming on others (e.g., aerial manipulation). To address this, we decompose the complex mission into distinct sub-tasks (derived from different starting position of robot in the whole terrain), assigning a dedicated single-configuration primitive to each. This isolation allows each primitive to specialize in the design and control required for a specific physical domain without interference from conflicting gradients.

## 4.1. Multi-tail Co-optimization Network Structure

Considering a base robot (raw input) equipped with $J$ independent hinge joints, we define a specific configuration as a derived state where a subset of these $J$ joints are rigidly locked, functioning as fixed structural connectors. Therefore, a configuration $C$ can be uniquely defined by two parameterized vectors. One is lock mask ($\mathbf{m} \in \{0,1\}^J$), a binary vector determining the structure of the configuration. For the $j$-th joint, $m_j = 1$ indicates that the joint is locked. The other is lock angle ($\mathbf{q}^{lock} \in \mathbb{R}^J$), a vector specifying the angle of the locked joints. If $m_j = 1$, the $j$-th joint is fixed at the angle $q_j^{lock}$. For active joints ($m_j = 0$), the corresponding value is ignored. Thus, a configuration can be denoted as a tuple $C = (\mathbf{m}, \mathbf{q}^{lock})$.

To match the distinct components of the configuration, we disentangle the complex gradient flow between design and control through a multi-tail policy architecture consisting of design and control (visualized in Fig. 2). Based on the design encoding, these can be further decomposed into three specialized sub-policies: the lock mask tail, lock angle tail, and control tail.

The architecture begins with a shared feature extractor. The extractor selects, aligns, and normalizes the possible input observations $O$, including the sub-task observation $o_t$ (e.g., height field of terrain), design observation $o_d$ $(\mathbf{m}, \mathbf{q}^{lock})$ and simulation observation $o_s$ (e.g., position and velocity), with output latent feature $h_{shared}$. Following the extractor, the latent feature $h_{shared}$ branches into three distinct tails, each responsible for a specific aspect of the robot's behavior:

**Lock mask tail** ($\pi_{mask}$)    This tail outputs the discrete probability distribution for the binary lock mask $\mathbf{m}$, determining the structural lock mask of the robot.

**Lock angle tail** ($\pi_{qpos}$)    This tail outputs the continuous distribution for the locking angles $\mathbf{q}^{lock}$, determining the specific geometric lock angle for the joints selected by the mask tail.

**Control tail** ($\pi_{ctrl}$)    This tail outputs the continuous motor commands (torque or position targets) for the active joints, driving the robot to execute the task dynamics.

To facilitate stable optimization via reinforcement learning, each of the three tails is further composed of an actor network and a critic network. The actor network maps the features to the specific action distribution (Bernoulli for the mask, Gaussian for angle and control), while the critic network estimates the value function $V(s)$ specific to that tail's objective.

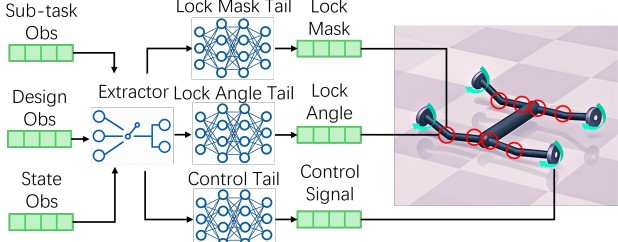

*Figure 2.* Illustration of the multi-tail structure in the single-configuration primitive. The observations (sub-task, design and state) are fed into the extractor, and then send to three tails which output lock mask, lock angle and control signal respectively. These design and control actions are used to define the configuration and drive the robot.

## 4.2. Co-optimization Primitive Training Process

We employ Proximal Policy Optimization (PPO) (Schulman et al., 2017) to optimize our multi-tail network. Reflecting the sequential dependency of our architecture, the inference and training procedure within each episode are formulated into three sequential stages.

**Stage 1: Lock Mask Decision**    The process initiates with the sub-task observation $o_t$. The extractor directly feeds $o_t$ to the lock mask tail policy $\pi_{mask}$, which outputs a Bernoulli distribution for each joint $j \in \{1, \ldots, J\}$, determining whether the joint should be locked ($m_j = 1$) or actuatable ($m_j = 0$). The generation of the binary lock mask vector $\mathbf{m}$ can be formulated as:

$$\mathbf{m} \sim \pi_{mask}(\mathbf{m}|o_t) \triangleq \prod_{j=1}^{J} \pi_{mask}(m_j|o_t) \qquad (2)$$

During our experiments, we observe that the optimization of design part sometimes leads the policy to converge prematurely to local optima. To mitigate this problem, we introduce an $\epsilon$-greedy exploration strategy specifically for design tails. More concretely, instead of relying solely on the distributions predicted by the actor networks, we force the design actions $\mathbf{m}$ to be sampled uniformly from the valid action space with a small probability $\epsilon$.

**Stage 2: Lock Angle Decision**    Conditioned on the determined lock mask, the system proceeds to determine the lock angle. The extractor augments the input by concatenating the $o_t$ with $o_d$ (where $o_d = \mathbf{m}$). The lock angle tail $\pi_{qpos}$ then samples the locking angles $\mathbf{q}^{lock}$ for the joints specified by the mask:

$$\mathbf{q}^{lock} \sim \pi_{qpos}(\mathbf{q}^{lock}|o_t, \mathbf{m}) \qquad (3)$$

This step finalizes the robot's configuration design $C = (\mathbf{m}, \mathbf{q}^{lock})$. Similarly, we apply the $\epsilon$-greedy exploration on $\mathbf{q}^{lock}$. Besides, we also find that simultaneously optimizing

both m and $\mathbf{q}^{lock}$ from scratch is sometimes unstable. To address this, we freeze the parameters of $\mathbf{q}^{lock}$ tail during the initial several steps, and unfreeze it later to fine-tune $\mathbf{q}^{lock}$.

**Stage 3: Dynamic Control**   With the robot's configuration fully instantiated, the system enters the continuous control loop. The input space is expanded to $o_t$, $o_d$ (where $o_d = C$), and $o_s$ (including the robot's proprioceptive state and surrounding terrain information). Notably, to stabilize training, the extractor applies normalization to $o_s$ before concatenating it with $o_t$ and $o_d$. The control tail $\pi_{ctrl}$ maps this unified context to continuous control signals $\mathbf{a}_t$ at each timestep $t$:

$$\mathbf{a}_t \sim \pi_{ctrl}(\mathbf{a}_t|o_s, o_t, \mathbf{m}, \mathbf{q}^{lock}) \qquad (4)$$

**Trajectory Formulation and Optimization**   To bridge the hierarchical inference process with PPO training, we structure the data collection and optimization objectives based on the temporal scope of each decision. For the control policy $\pi_{ctrl}$, we collect standard step-wise trajectories $\tau_{ctrl} = \{(s_t, \mathbf{a}_t, r_t, s_{t+1})\}_{t=0}^T$, where the state $s_t$ represents the unified context vector $[o_t, C, o_s]$ and $r_t$ is the immediate reward from the environment at timestep $t$. For most tasks, $r_t$ mainly consists of the forward locomotion item, and for some tasks requiring up-down moving (e.g. mountain terrain), we add an auxiliary up-down locomotion item.

For the design policies ($\pi_{mask}$ and $\pi_{qpos}$), optimizing the one-shot configuration decisions requires addressing the delayed feedback loop where an initial choice influences the entire episode. To address this, we adopt the temporal credit assignment mechanism from BodyGen (Lu et al., 2025), formulating the design interaction as a single-step transition: the "state" is the initial task observation $o_t$, the "action" is the generated configuration $C = (\mathbf{m}, \mathbf{q}^{lock})$, and the "reward" is the cumulative episodic return $R = \sum_{t=0}^T \gamma^t r_t$. Consequently, the design value networks are trained to estimate the global expected return $L_{design} \approx (R - V_{design}(o_t))^2$, ensuring the morphology is optimized for long-horizon task success.

## 5. Multi-Configuration Scheduler

Having established the methodology for co-optimization primitives for individual configurations in the previous section, we now address the challenge of integrating these primitives into a unified system. In this section, we present the multi-configuration scheduler, a hierarchical module designed to schedule $K$ pre-trained primitives to accomplish complex tasks.

### 5.1. Configurations Scheduler Structure

Integrating configuration scheduler and motion control into a flat policy proves ineffective due to their conflicting scopes. The former requires global awareness to plan long-horizon adaptations, whereas the latter demands local, high-frequency reactions to terrain features. Therefore, we use temporal abstraction through a hierarchical structure to decouple these responsibilities: a high-level scheduler handles global configuration planning among the previous pre-trained primitives, solving the temporal credit assignment problem that end-to-end methods fail. For the $K$ pre-trained primitives, we first freeze their parameters and infer their design tails (lock mask and angle tails) to generate $K$ deterministic configurations. Let $C_i = (\mathbf{m}_i, \mathbf{q}_i^{lock})$ denote the configuration derived from the $i$-th primitive. These fixed configurations form a discrete action space $\{C_i\}_{i=1}^K$. The scheduler relies on the following components.

**Scheduler Network**   A high-level policy $\pi_{dec}$ parameterized by an MLP. It takes the global observation $o_{global}$ (containing the global task information and the simulation observation from the environment) as input and outputs a categorical distribution over $\{C_i\}_{i=1}^K$, selecting an active configuration index every $M$ simulation steps.

**Transition Controller**   To bridge the kinematic gap between different configurations, we implement a simple PD controller. The PD controller shares the same parameters for all tasks with similar physical properties. When the scheduler switches the target, this controller is activated to drive the robot's joints to the target locking angles before the new primitive takes control.

### 5.2. Decider Training Process

In this stage, we freeze all pre-trained primitives and optimize the scheduler policy $\pi_{dec}$ via PPO. The execution is segmented into intervals of $M$ steps. At the start of each interval, the scheduler samples a target configuration $C_{target}$ based on the global observation. If $C_{target}$ differs from the current configuration $C_{curr}$, the transition controller is triggered, and a reconfiguration penalty is applied to the reward function to discourage frequent switching. Then the total reward is $R = R_{locomotion} - R_{penalty}$. Subsequently, the control tail of the selected primitive $C_{target}$ takes over to drive the robot for the remainder of the interval.

## 6. Results

### 6.1. Experiments Setup

**Task Description**   We comprehensively evaluate our hierarchical co-optimization framework on a diverse suite of multi-physics locomotion tasks. The implementation de-

*Table 1.* Comparison of normalized $S$ with the single-robot algorithms. Higher value means better performance. This table reports the medium value and the standard deviation across eight different random seeds.

| Algorithms | Fly-walk (mount) | Fly-walk (pit) | Car-leg (stairs) | Car-leg (narrow) |
|---|---|---|---|---|
| Ours | **1.978** $\pm$ 0.117 | **1.417** $\pm$ 0.102 | **0.872** $\pm$ 0.099 | **2.618** $\pm$ 0.125 |
| Domain-Expert | 1.668 $\pm$ 0.089 | 0.895 $\pm$ 0.025 | 0.626 $\pm$ 0.031 | 1.050 $\pm$ 0.028 |
| PPO | 0.403 $\pm$ 0.018 | 0.343 $\pm$ 0.020 | 0.508 $\pm$ 0.044 | 2.318 $\pm$ 0.092 |
| TD-MPC2 | 0.219 $\pm$ 0.011 | 0.318 $\pm$ 0.033 | 0.519 $\pm$ 0.058 | 0.508 $\pm$ 0.025 |

| Algorithms | Wheel-leg | Wheel-fly (mount) | Wheel-fly (swim) |
|---|---|---|---|
| Ours | **9.148** $\pm$ 0.274 | **7.368** $\pm$ 0.201 | **3.967** $\pm$ 0.231 |
| Domain-Expert | 8.860 $\pm$ 0.101 | 6.310 $\pm$ 0.098 | 3.664 $\pm$ 0.037 |
| PPO | 0.210 $\pm$ 0.031 | 0.453 $\pm$ 0.072 | 0.346 $\pm$ 0.060 |
| TD-MPC2 | 0.236 $\pm$ 0.016 | 0.498 $\pm$ 0.047 | 0.259 $\pm$ 0.056 |

tails and hyperparameters of these experiments are listed in App. A. Each task is defined by a pair consisting of a raw robot embodiment and a complex terrain (App. B.1). Crucially, these terrains are constructed as sequences of heterogeneous sub-terrains (e.g., flat ground, gaps, steps, high mountain, or water), representing distinct sub-tasks that demand specific locomotion ability for walking, flying or swimming. Taking the raw robot and the terrain map as input, our goal is to discover an adaptive policy that co-optimizes a set of feasible reconfiguration designs and their corresponding controllers, enabling the robot to efficiently traverse these challenging multi-physics environments.

**Baselines** Since the problem we proposed (co-optimizing of reconfigurable robots) is novel and no existing work fits these tasks, we evaluate our method against two categories of state-of-the-art baselines to answer the following questions: whether reconfiguration in control policy is necessary; whether auto-designed configuration can beat human intuition; whether existing co-design algorithms can adapt to these heterogeneous tasks.

The first category (Sec. 6.2) focuses on control algorithms applied to the same raw robot embodiment, testing the necessity of our automated reconfiguration mechanism. Specifically, we apply two reinforcement learning algorithms, including **PPO** (Schulman et al., 2017) and **TD-MPC2** (Hansen et al., 2023) directly to the raw robot without any reconfiguration structure (i.e., all joints are active), determining whether sophisticated control alone suffices for these complex tasks. Additionally, we evaluate **Domain-expert** design configurations, where we manually design fixed configurations based on human intuition for specific sub-terrains, assessing whether our automated design policy performs better than human expert knowledge.

The second category (Sec. 6.3) consists of co-design algorithms that optimize both robot embodiments and control given only the task environment, testing the capability of current SOTA methods on our complex tasks. We benchmark

against representative frameworks including **BodyGen** (Lu et al., 2025) and **GLSO** (Hu et al., 2023) to investigate whether existing co-design methods can effectively generate embodiments capable of traversing the long-horizon, heterogeneous terrains presented in our experiments.

**Metrics** We evaluate the locomotion performance on our terrain traversal tasks using the normalized traversal progress $S$. Specifically, we measure the forward distance $d_{agent}$ traveled by the robot within a fixed time horizon, normalized by the length of the primary challenge section of the map, $L_{map}$. Therefore, the metric is defined as $S = d_{agent}/L_{map}$. Note that the terrain extends infinitely with the final sub-terrain pattern beyond $L_{map}$ to accommodate successful agents ($S > 1$).

### 6.2. Comparison with Single-Robot Algorithms

We report the quantitative comparison results with the single-robot control algorithms in Tab. 1 and qualitative visualizations in App. B.2, with videos in supplementary materials. On average, our hierarchical co-optimization framework significantly outperforms all single-robot baselines across the diverse set of evaluation tasks (1.19x Domain-expert, 5.97x PPO, and 10.70x TD-MPC2), demonstrating the critical necessity of adaptive reconfiguration.

**Analysis of reconfiguration-free baselines.** The performance of PPO and TD-MPC2 exhibits the limitation of reconfiguration-free control policies in complex environments. As shown in the table, these baselines typically achieve normalized scores well below 1.0, a failure to complete the full terrain, and indicate convergence to local optima tailored to a specific sub-task. Due to the conflicting physical requirements of heterogeneous terrains (e.g., only walk ability fails to drive the robot in water), a fixed policy often overfits to the robot's initial or dominant capabilities, mastering one sub-task while incapable of traversing the subsequent distinct terrain. On the other hand, our

*Table 2.* Comparison of normalized $S$ with the multi-morphology algorithms. Higher value means better performance. "Scratch" means training from scratch, and "init" means training from the same initial morphology as ours.

| ALGORITHMS | FLY-WALK (MOUNT) | FLY-WALK (PIT) | CAR-LEG (STAIRS) | CAR-LEG (NARROW) | WHEEL-LEG | WHEEL-FLY (MOUNT) | WHEEL-FLY (SWIM) |
|---|---|---|---|---|---|---|---|
| OURS | **1.978** | **1.417** | **0.872** | **2.618** | **9.148** | **7.368** | **3.967** |
| BODYGEN (SCRATCH) | 0.498 | 0.372 | 0.453 | 0.436 | FAIL | 0.498 | 0.256 |
| GLSO (SCRATCH) | 0.507 | 0.364 | 0.505 | 0.536 | 0.251 | 0.507 | 0.341 |
| BODYGEN (INIT) | 0.452 | 0.443 | 0.511 | 0.484 | FAIL | 0.515 | 0.293 |
| GLSO (INIT) | 0.496 | 0.483 | 0.367 | 0.535 | 0.264 | 0.502 | 0.355 |

*Table 3.* Ablation study with deleting $\epsilon$-greedy and stage optimization modules respectively. Higher value means better performance.

| ALGORITHMS | FLY-WALK (MOUNT) | FLY-WALK (PIT) | CAR-LEG (STAIRS) | CAR-LEG (NARROW) | WHEEL-LEG | WHEEL-FLY (MOUNT) | WHEEL-FLY (SWIM) |
|---|---|---|---|---|---|---|---|
| FULL PIPELINE | **1.978** | **1.417** | **0.872** | **2.618** | **9.148** | **7.368** | **3.967** |
| W/O $\epsilon$-GREEDY | 0.335 | 0.230 | 0.651 | 1.567 | 8.923 | 3.820 | 3.400 |
| W/O STAGE OPT | 0.437 | 0.300 | 0.550 | 1.441 | 8.723 | 0.325 | 3.033 |

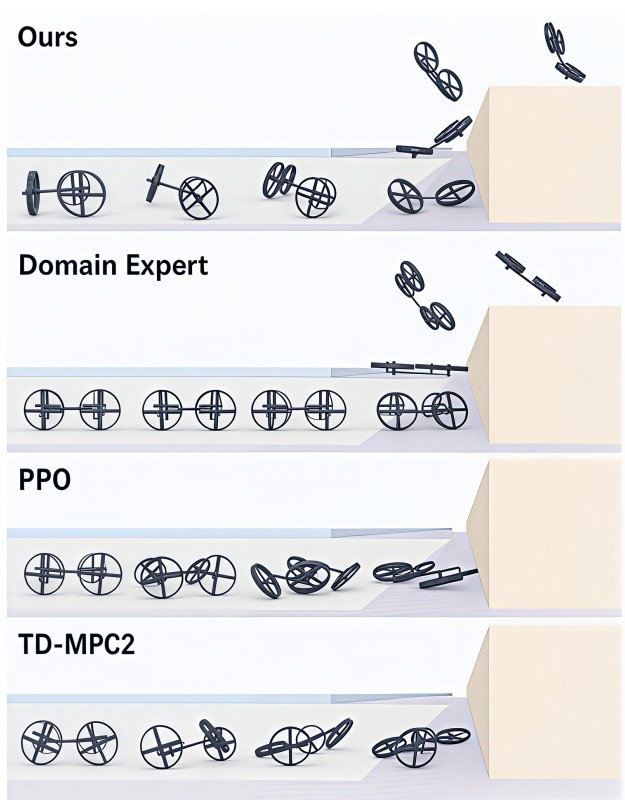

*Figure 3.* Visualization of the comparison among our pipeline, Domain-expert, PPO, and TD-MPC2 on *wheel-fly (mount)* task. PPO and TD-MPC2 are all stuck at the bottom of the mountain. While domain-expert successfully fly over the mountain, our pipeline demonstrate higher performance.

method dynamically adapts the configurations based on the current physical requirements, satisfying spatially varying constraints. Specifically, visual inspection of the *Wheel-fly (Mount)* experiment in Fig. 3 reveals that our agent discovers physically specialized configurations, combining the characteristics of wheel and fly to switch between crossing the tunnel and flying over the mountain, while the baselines struggle between these terrains and stop before the mountain.

**Advantages over domain experts.** The domain-expert baselines based on long-existing classical structures in robotics communities generally succeed in completing the tasks ($S > 1.0$), validating that reconfiguration is indeed the key to solving these long-horizon challenges. However, our automated method consistently surpasses these human-engineered solutions by optimizing the lock mask and lock angle beyond human intuition. Also in the task *Wheel-fly (Mount)* (Fig. 3), our method finds a more efficient configuration, where the robot turns its back wheels forward to propel it in the tunnel rather than purely working as a four-wheel car. This superiority stems from our design heads' ability to optimize beyond broad structural changes, discovering fine-grained geometric advantages—such as precise joint locking angles—that maximize task efficiency but are difficult for human designers to intuit manually.

### 6.3. Comparison with Morphology-Co-Design Algorithms

We report the quantitative comparison with the morphology co-design algorithms (with only terrain input and output the robot embodiments as well) in Tab. 2 and the qualitative visualizations in App. B.3, with videos in supplementary materials. We use these baselines to demonstrate the state-

of-the-art co-design (from scratch and from the same initial morphology as ours) methods for the heterogeneous tasks. The results exhibit that while these baselines successfully generate functional robot embodiments and corresponding control policies optimized for specific terrain traits, they still fail to adapt to the complex tasks and are surpassed by our pipeline (10.52x BodyGen and 9.10x GLSO on average). Both of them generally fail to complete the long-horizon tasks, with normalized scores frequently hovering around 0.5. This underperformance stems from their formulation: these methods typically seek a single, globally optimal morphology for the entire episode. However, in our highly heterogeneous environments, no such "master key" morphology exists, and therefore these algorithms are forced to converge to a compromise design that performs mediocrely across sub-tasks or overfit to the first terrain segment, leading to early failure when the terrain features shift. For example, as visualized in Fig. 4, the baselines output morphologies efficient for the flat plane while physically incapable of traversing rough steps (stuck behind the stairs). In contrast, our hierarchical framework achieves scores significantly exceeding 1.0 (e.g., 9.148 in *Wheel-leg*), demonstrating successful and repeated traversal of the entire map. Also referring to Fig. 4, our pipeline switches between car and leg configurations, adapting to both plane and stairs. This validates that for complex, multi-physics tasks, the ability to dynamically reconfigure between multiple configurations is a prerequisite for success. Our method's capacity to deploy specialized executors for each terrain segment allows it to bypass the morphological trade-offs that constrain static-embodiment approaches.

### 6.4. Ablation Studies

To validate the effectiveness of our hierarchical pipeline and the specific mechanisms for mitigating local optima (Sec. 4.2), we conduct two ablation studies. First, we compare our pipeline with an end-to-end algorithm that combines the primitives and scheduler in one policy and simultaneously optimizes them and a heuristic method which replaces our learned scheduler with a rule-based heuristic scheduler. Second, we remove the $\epsilon$-greedy exploration strategy and the staged optimization mechanism (mentioned in Sec. 4.2), respectively. The quantitative results are summarized in Fig. 5 and Tab. 3, and the qualitative visualizations are shown in App. B.4. Videos are provided in supplementary materials. Our pipeline achieves 9.83x, 1.47x and 1.80x performance boost over the end-to-end, w/o $\epsilon$-greedy and w/o stage optimization method respectively.

Regarding the end-to-end method, our hierarchical pipeline significantly outperforms them across all evaluated tasks. The end-to-end method exhibits poor performance in these complex heterogeneous environments, where they only achieve parts of the whole task with $S$ hovering under 0.5,

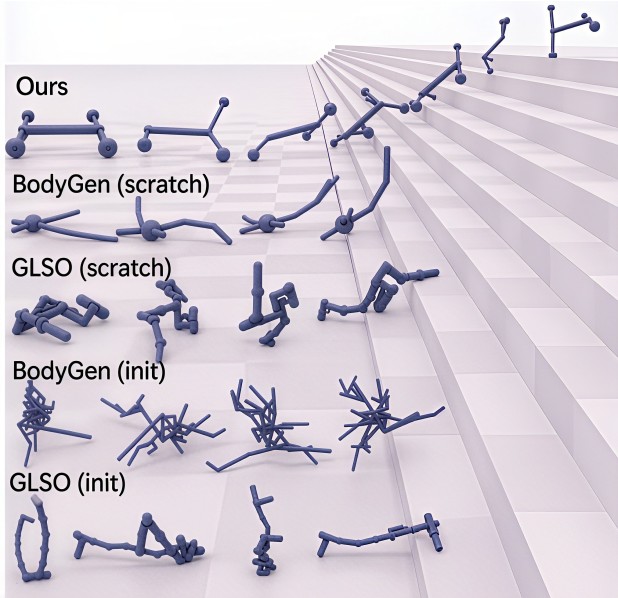

*Figure 4.* Visualization of the comparison among our pipeline, BodyGen, and GLSO on *car-leg (stairs)* task. While BodyGen and GLSO are all stuck behind the stairs, our pipeline successfully traverse upstairs.

while our methods reach remarkable higher scores. This substantial performance gap suggests that simultaneously optimizing high-level scheduler and low-level primitives in a single policy creates a conflicting gradient landscape. Without the explicit temporal decomposition provided by our hierarchy, the end-to-end agent struggles to balance the long-horizon planning required for reconfiguration with the precise local control needed for locomotion, leading to sub-optimal convergence.

Meanwhile, the heuristic approaches, as shown in the results, are proved to be highly ineffective in most cases. Correct switching timing cannot rely on simple distance or position rules; it requires complex physical state fusion (e.g., comprehensively evaluating qpos and qvel across all joints to account for dynamic braking distances). Our learned scheduler is essential because it robustly processes these high-dimensional dynamics to ensure stable transitions.

On the other hand, removing the $\epsilon$-greedy strategy ("w/o $\epsilon$-greedy") also leads to a drastic score drop in certain environments like *Fly-walk (mount)* ($1.978 \rightarrow 0.335$), confirming that without forced exploration, the policy tends to converge prematurely to sub-optimal topologies. Likewise, the removal of the staged optimization mechanism ("w/o stage optimization") also results in failure on some tasks. For instance, in *Wheel-fly (mount)*, the score collapses from 7.368 to 0.325, showing significance of the warm-up phase for the lock mask head during training. Although these influences are more mild on other tasks, the widespread degradation on the removal of $\epsilon$-greedy strategy and staged optimization

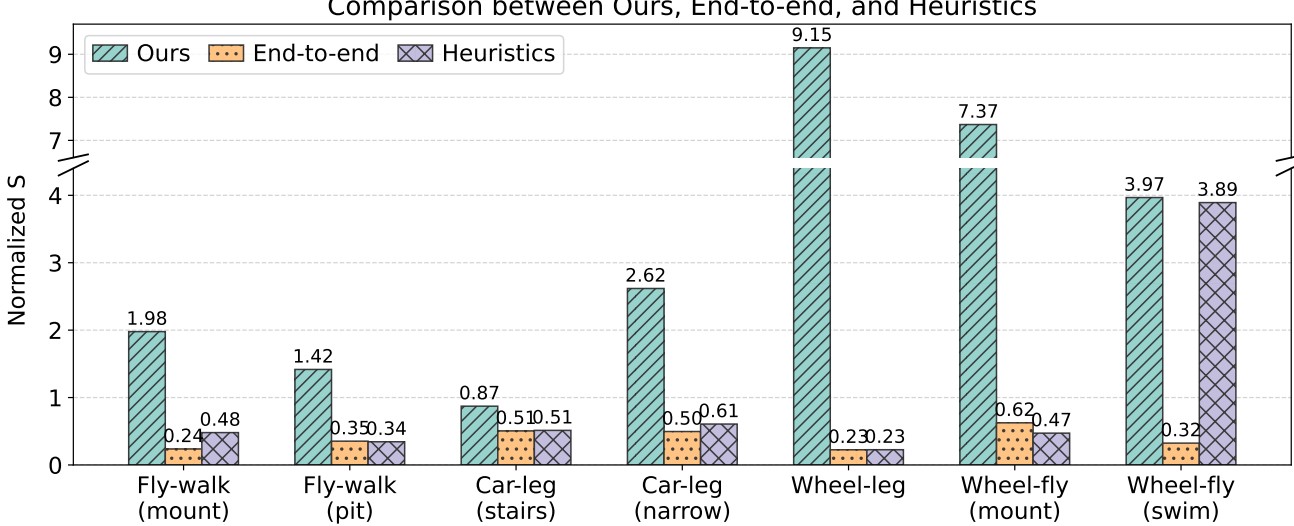

*Figure 5.* Comparison of normalized $S$ between our method (green), end-to-end method (orange) and heuristic method (purple).

emphasizes that the introduction of these two modules can effectively avoid convergence to local optimum.

## 7. Conclusions and Limitations

In this work, we present a hierarchical reconfiguration co-design framework capable of automating the generation of an integrated policy on a given robot embodiment and a long-horizon, heterogeneous task in multi-physics environments, including terrestrial, aerial, and aquatic physics. We decouple the problem into primitive exploration of design and control for each configuration with a multi-tail primitive and a global scheduler to switch these primitives. The experiments demonstrate that our pipeline outperforms the state-of-the-art single-robot control baselines and reconfigurable-free co-design methods, validating that dynamic reconfiguration is a prerequisite for robotic versatility in complex worlds.

Despite these advancements, we acknowledge several limitations that point towards important future directions. First, our pipeline now depends on pre-defined robot embodiments as input to derive the reconfigurable configurations. Future works can integrate the optimization of the robot embodiment into the pipeline. Second, our reward function is designed primarily to optimize task completion and locomotion speed, neglecting the energy efficiency of the system. Future works can try to balance the energy cost and locomotion speed. Third, the current pipeline does not explicitly account for physical manufacturing requirements, which can be incorporated as constraints in the future.

## Acknowledgments

We would like to thank Tsinghua University, Shanghai Qi Zhi Institute, Xi'an Jiaotong University for their financial support throughout this research. Tao Du acknowledges the support from Tsinghua University and the Shanghai Qi Zhi Institute Innovation Program.

## Impact Statement

This paper presents work whose goal is to advance the field of machine learning and robotics. The proposed method for reconfigurable robot co-optimization has many potential societal consequences, particularly in enhancing the versatility of autonomous systems in complex environments. While we believe the primary impact will be enabling more efficient robotic assistance and exploration, there are no specific negative ethical consequences which we feel must be specifically highlighted here beyond the standard considerations for autonomous systems.

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

# A. Details of Experiments

## A.1. Implementation Details

We implement the simulation environment based on python and MuJoCo with OpenGL for results visualization. We build our policy network structure via PyTorch, combining with Stable Baselines 3 for the training process of reinforcement learning. We evaluate our pipeline on a server with an AMD EPYC 9754 128-Core CPU, $12\times$ DDR5 4800 16GB (384GB in total) RAM, and $8 \times$ NVIDIA RTX 4090 24G GPU.

## A.2. Hyper-Parameters

**Simulation Parameters**  Tab. 4 details the simulation parameters, underscoring the multi-physics breadth of our experimental suite. The environments encompass distinct physical regimes, ranging from high-viscosity hydrodynamics (Water Density $1000\,\mathrm{kg/m^3}$) to low-density aerodynamics (Air Density $1.225\,\mathrm{kg/m^3}$), alongside standard rigid-body contact dynamics. Furthermore, the variation in numerical integrators (RK4 vs. ImplicitFast) and simulation timesteps ($1/250$ ms to $1/1500$ ms) demonstrates that our pipeline is robust across widely varying dynamic stiffness and temporal resolutions, validating the robot's adaptability to fundamentally different physical laws.

*Table 4.* Simulation Parameters of each Experiment.

| PARAMETERS | FLY-WALK (MOUNT) | FLY-WALK (PIT) | CAR-LEG (STAIRS) | CAR-LEG (NARROW) |
|---|---|---|---|---|
| TIMESTEP (S) | 1/250 | 1/250 | 1/250 | 1/250 |
| INTEGRATOR | RK4 | RK4 | RK4 | RK4 |
| ROBOT DENSITY $(\mathrm{kg/m^3})$ | 600 | 600 | 1000 | 1000 |
| AIR DENSITY $(\mathrm{kg/m^3})$ | 1.225 | 1.225 | N/A | N/A |
| AIR VISCOSITY $(\mathrm{Pa \cdot s})$ | $1.5 \times 10^{-5}$ | $1.5 \times 10^{-5}$ | N/A | N/A |
| WATER DENSITY $(\mathrm{kg/m^3})$ | N/A | N/A | N/A | N/A |
| WATER VISCOSITY $(\mathrm{Pa \cdot s})$ | N/A | N/A | N/A | N/A |

| PARAMETERS | WHEEL-LEG | WHEEL-FLY (MOUNT) | WHEEL-FLY (SWIM) | |
|---|---|---|---|---|
| TIMESTEP (S) | 1/1500 | 1/250 | 1/250 | |
| INTEGRATOR | IMPLICITFAST | RK4 | RK4 | |
| ROBOT DENSITY $(\mathrm{kg/m^3})$ | 1100 | 1000 | 2000 | |
| AIR DENSITY $(\mathrm{kg/m^3})$ | N/A | 1.225 | N/A | |
| AIR VISCOSITY $(\mathrm{Pa \cdot s})$ | N/A | $1.5 \times 10^{-5}$ | N/A | |
| WATER DENSITY $(\mathrm{kg/m^3})$ | N/A | N/A | 1000 | |
| WATER VISCOSITY $(\mathrm{Pa \cdot s})$ | N/A | N/A | $1 \times 10^{-3}$ | |

**Network Structure**  Tab. 5 details the specific architectures used in our pipeline. We employ moderately sized MLPs in our pipeline. This design choice results in a minimal GPU memory footprint (under 500 MB), ensuring the entire pipeline remains lightweight and flexible. This compact architecture demonstrates the potential for deployment on resource-constrained onboard hardware without sacrificing performance.

*Table 5.* Network Structure of each Policy.

| COMPONENTS | LOCK-MASK HEAD | LOCK-QPOS HEAD | CONTROL HEAD | SCHEDULER |
|---|---|---|---|---|
| ACTOR MLP | $512 \times 512 \times 512$ | $512 \times 512 \times 512$ | $1024 \times 1024 \times 1024$ | $1024 \times 1024$ |
| CRITIC MLP | $512 \times 512 \times 512$ | $512 \times 512 \times 512$ | $1024 \times 1024 \times 1024$ | $1024 \times 1024$ |
| ACTIVATION FUNCTION | ELU | ELU | ELU | RELU |

**Training Parameters**  Tab. 6 lists the primary training hyperparameters used across our experiments. For $\epsilon$-greedy rate and stage optimization start steps, please refer to Sec. 5.1. Empirically, we observe that our pipeline exhibits robustness to hyperparameter variations. The system remains effective across a reasonable range of values rather than requiring precise, brittle tuning for each specific task. This stability indicates that the method is not overly sensitive to specific configurations, facilitating easier reproduction and adaptation to new environments.

*Table 6.* Major training parameters of each experiment. $A \to B$ means linearly descending from $A$ to $B$.

| PARAMETERS | FLY-WALK (MOUNT) | FLY-WALK (PIT) | CAR-LEG (STAIRS) | CAR-LEG (NARROW) |
|---|---|---|---|---|
| LEARNING RATE | $3 \times 10^{-4}$ | $3 \times 10^{-4}$ | $3 \times 10^{-4}$ | $4 \times 10^{-4}$ |
| ENTROPY COEFFICIENT | 0.04 | 0.05 | 0.02 | 0.02 |
| $\epsilon$-GREEDY RATE | $0.05 \to 0.01$ | $0.04 \to 0.01$ | $0.05 \to 0.01$ | $0.05 \to 0.01$ |
| STAGE OPT START STEPS | 30M | 30M | 30M | 30M |

| PARAMETERS | WHEEL-LEG | WHEEL-FLY (MOUNT) | WHEEL-FLY (SWIM) | |
|---|---|---|---|---|
| LEARNING RATE | $3 \times 10^{-4}$ | $3 \times 10^{-4}$ | $3 \times 10^{-4}$ | |
| ENTROPY COEFFICIENT | 0.02 | 0.05 | 0.02 | |
| $\epsilon$-GREEDY RATE | $0.05 \to 0.01$ | $0.04 \to 0.01$ | $0.05 \to 0.01$ | |
| STAGE OPT START STEPS | 30M | 40M | 30M | |

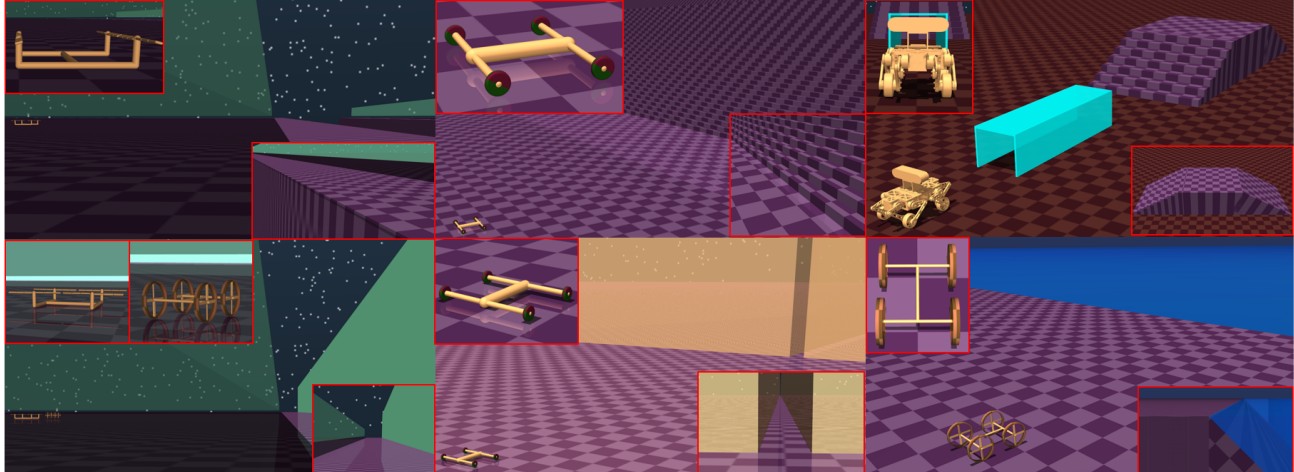

*Figure 6.* The full gallery of our tasks. Left-top: *fly-walk (pit)* task, consisting of two tunnels and a wide deep pit between them; Middle-top: *car-leg (stairs)* task, consisting of alternating plane and stairs; Right-top: *wheel-leg* task, consisting of tunnel, stairs and slope; Left-bottom: *fly-walk (mount)* and *wheel-fly (mount)*: consisting of a tunnel and a high mountain; Middle-bottom: *car-leg (narrow)* task, consisting of plane and narrow lane; Right-bottom: *wheel-fly (swim)* task, consisting of plane and water.

## B. Additional Results of Experiments

### B.1. Full Gallery of Robot Configurations

We visualize the full gallery of our heterogeneous tasks in Fig. 6 and the robot configurations explored in our experiments in Fig. 7. The diverse suite of heterogeneous tasks serves as a rigorous stress test for robotic versatility. Unlike standard benchmarks that focus on single-domain locomotion, our environments impose drastic transitions between conflicting physical regimes within a single episode. For instance, the *Fly-walk (mount)* scenarios require the agent to seamlessly shift from friction-based terrestrial traction to aerodynamics. These sharp discontinuities in environmental dynamics create a high-complexity control landscape where static morphologies inevitably fail, thereby validating the necessity of the proposed reconfiguration capabilities. The robot gallery illustrates this high diversity and physical specialization of the configurations discovered by our pipeline. In terrestrial segments, the robot typically adopts stable, quadrupedal-like forms with high ground clearance to navigate stairs and uneven terrain (e.g., *Car-leg* and *Wheel-leg* tasks). On the other hand, for aerial segments (e.g. *Fly-walk*) tasks, the agent discovers aerodynamic configurations where limbs are retracted or extended symmetrically to balance the center of mass during flight. Besides, in aquatic environments (e.g., *Wheel-fly (swim)*), the robot utilized the propellers to swim in the water and compensate for the insufficient buoyancy. This morphological adaptation aligns with physical intuition: the system automatically trades off between the stability required for gravity-dominated locomotion and the hydrodynamic efficiency needed for fluid interaction, validating the effectiveness of our co-design framework in exploring the vast design space.

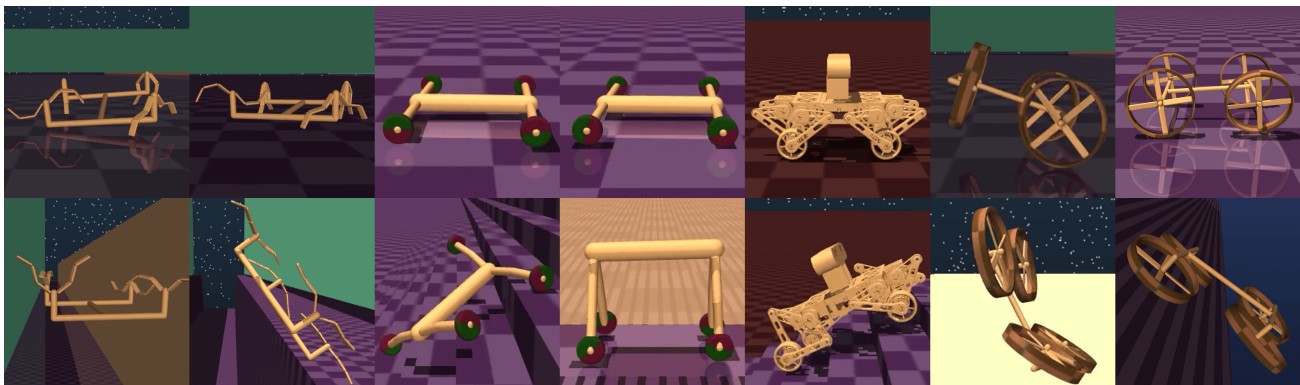

*Figure 7.* Full gallery of different robot configurations explored in our experiments. From left to right are fly-walk (mount), fly-walk (pit), car-leg (stairs), car-leg (narrow), wheel-leg, wheel-fly (mount), wheel-fly (swim) sequentially.

### B.2. Visualization of Comparison with Single-robot Control Baselines

We visualize the comparison with single-robot control baselines (manually-designed, PPO, and TD-MPC2) in Fig. 8 to Fig. 14. The visual sequences provide a qualitative breakdown of why static baselines fail. Across the tasks, we observe a consistent pattern: baseline policies typically master the initial sub-task (e.g., walking on flat ground) but suffer catastrophic failure at the terrain transition boundaries. For instance, in *Fly-walk (mount)*, static agents lacking aerodynamic configurations fail to generate sufficient lift over the mountain. Similarly, in *Wheel-fly (swim)*, they stall at the water's edge or sink due to excessive hydrodynamic drag. In contrast, our method is shown to execute timely reconfigurations—transforming from legged to aerial or aquatic forms—allowing it to seamlessly negotiate these physical discontinuities where fixed embodiments are physically incapable of proceeding.

On the other hand, while the domain-expert configurations generally succeed in completing basic traversal tasks, our experiments reveal that human intuition often falls short of optimality. Human designers typically rely on conservative, predefined templates (e.g., standard quadrupedal gaits or symmetric drone frames), which may not fully exploit the robot's kinematic redundancy. In contrast, our pipeline discovers non-intuitive yet highly efficient configurations that remain unexplored by human experts. For example, in the *Car-leg (stairs)* task, our method chooses to lock some of the leg joints instead of freeing all of them, remarkably increasing the locomotion performance on stairs. This demonstrates that our automated co-design framework not only matches expert performance but actively transcends it by navigating the design space more comprehensively.

### B.3. Visualization of Comparison with Morphology-Co-Design Algorithms

We visualize the comparison with morphology-co-design algorithms (BodyGen and GLSO) in Fig. 15 to Fig. 21. The visualizations highlight the fundamental limitation of traditional reconfiguration-free co-design algorithms in multi-physics environments: the inevitable trade-off between specialization and generalization. We observe that baselines like BodyGen and GLSO typically converge to "compromise morphologies" that attempt to satisfy conflicting physical requirements simultaneously. They often overfit to the initial terrain, carrying "morphological burdens"—such as heavy limbs required for walking—that become significant sources of drag or instability when the robot enters aquatic or aerial domains. In contrast, our hierarchical approach bypasses this dilemma entirely. By dynamically reconfiguring, the agent effectively "sheds" these burdens, deploying a sequence of highly specialized forms that fit each specific sub-task without being constrained by the needs of future or past terrains.

### B.4. Visualization of Ablation Studies

We visualize the comparison with the ablation studies in Fig. 22 to Fig. 28. The visual comparison with the end-to-end baseline reveals the behavioral consequences of coupling high-level planning with low-level control. Unlike our hierarchical approach, which is able to dynamically switch between configurations based on local environment, the end-to-end policy consists of a fixed scheduler and biased-trained primitives, leading to failure when physical properties change. This visual evidence corroborates that without the temporal abstraction provided by our hierarchy, the agent fails to learn the distinct causality between morphology changes and sub-task transitions.

On the other hand, visualizations of the ablation studies on the $\epsilon$-greedy and stage optimization strategies highlight how the proposed mechanisms prevent convergence to local optima. In the w/o $\epsilon$-greedy trials, the agent often settles for "lazy" configurations—generic shapes that are minimally viable for the start but physically incapable of traversing subsequent hard obstacles. Similarly, the w/o staged optimization variants display kinematic instability, where the robot attempts to control a topology that has not yet stabilized, leading to chaotic flailing rather than purposeful locomotion. These visual artifacts underscore that our structured exploration and warm-up phases are critical for discovering high-quality, specialized embodiments rather than just barely functional ones.

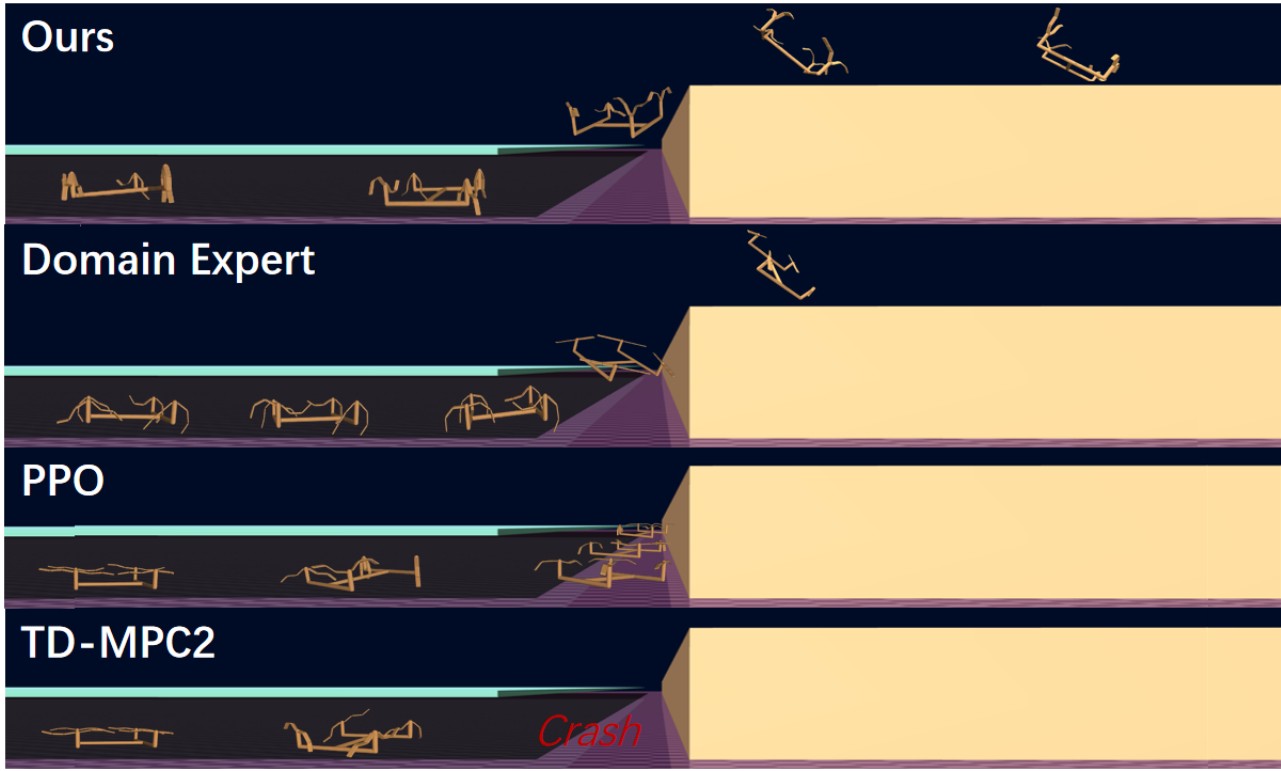

*Figure 8.* Visualization of comparison with single-robot baselines of Fly-walk (Mount) experiment, containing four frames in $T = 0, 5, 10, 15, 20$ s respectively. "Crash" means the algorithm makes the robot crash at this point.

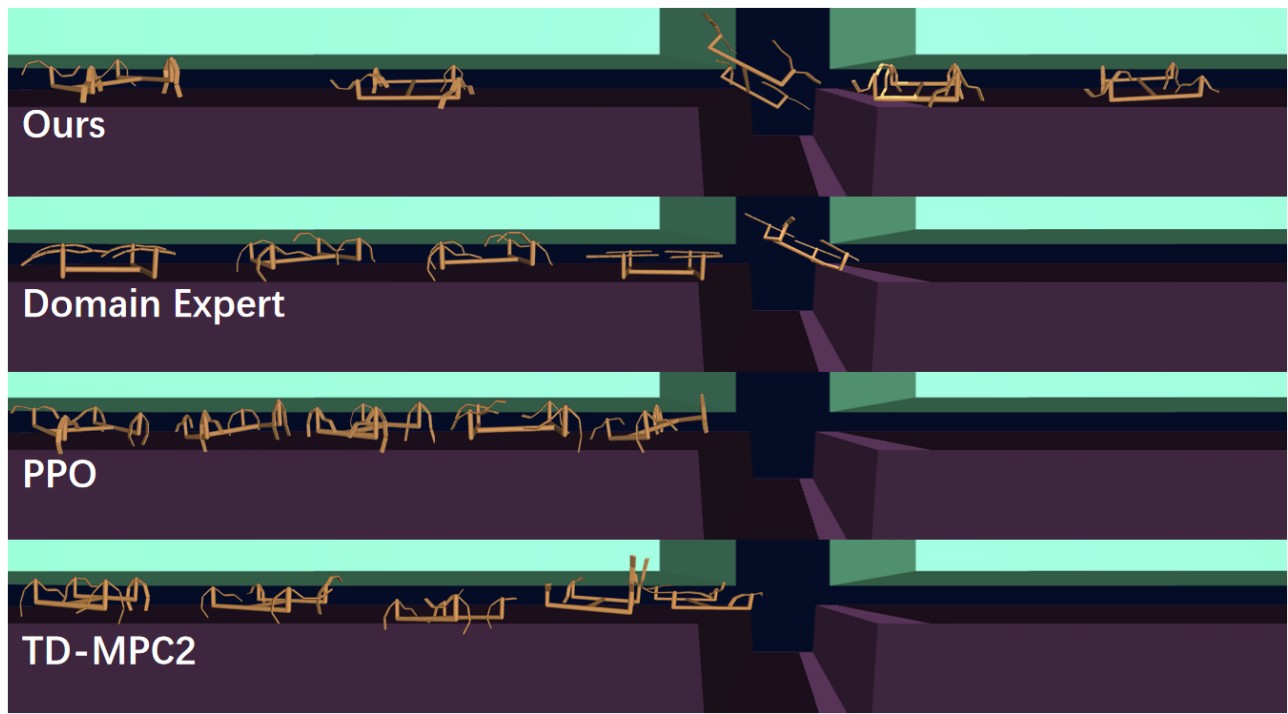

*Figure 9.* Visualization of comparison with single-robot baselines of Fly-walk (Pit) experiment, containing four frames in $T = 0, 5, 10, 15, 20$ s respectively.

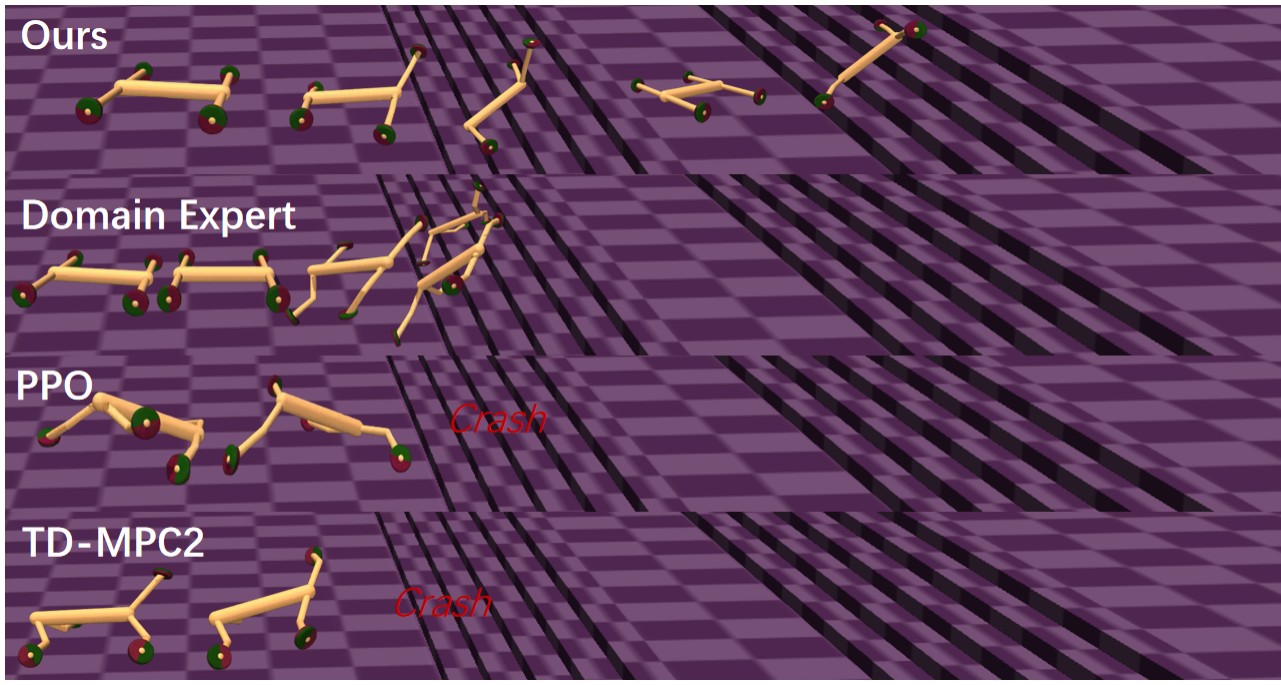

*Figure 10.* Visualization of comparison with single-robot baselines of Car-leg (Stairs) experiment, containing four frames in $T = 0, 7, 14, 21, 28$ s respectively. "Crash" means the algorithm makes the robot crash at this point.

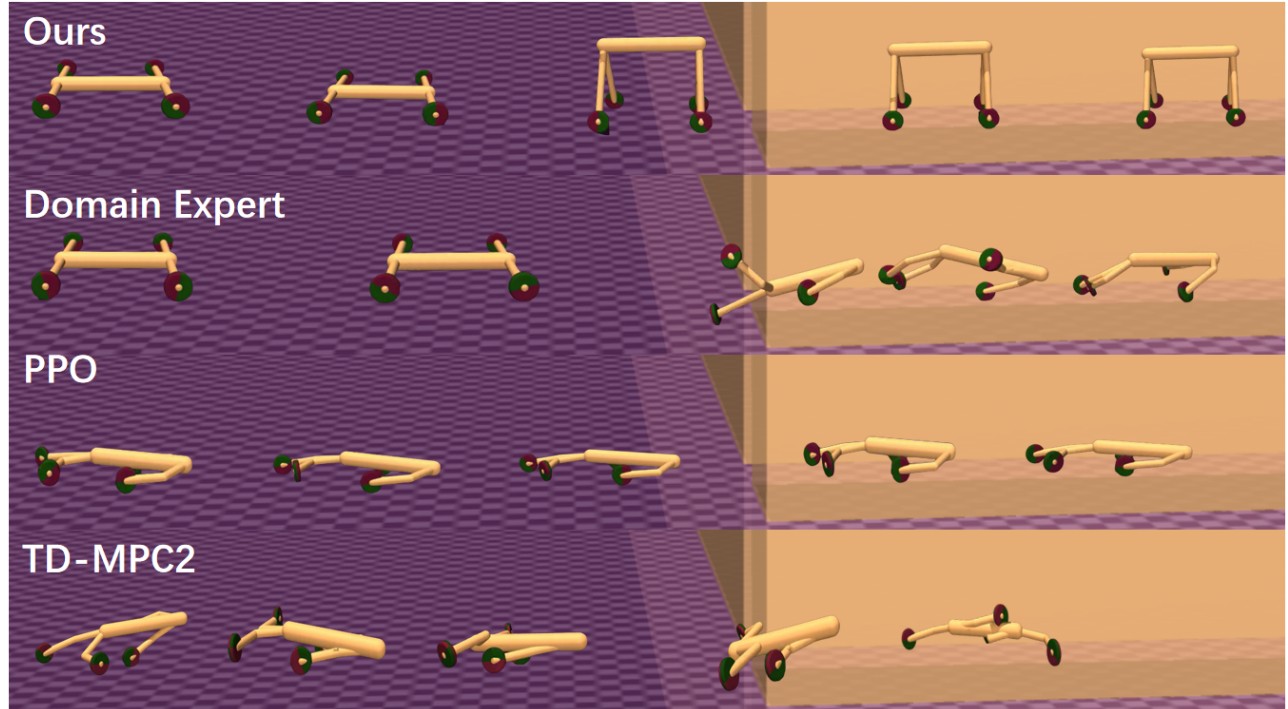

Figure 11. Visualization of comparison with single-robot baselines of Car-leg (Narrow) experiment, containing four frames in $T = 0, 5, 10, 15, 20$ s respectively.

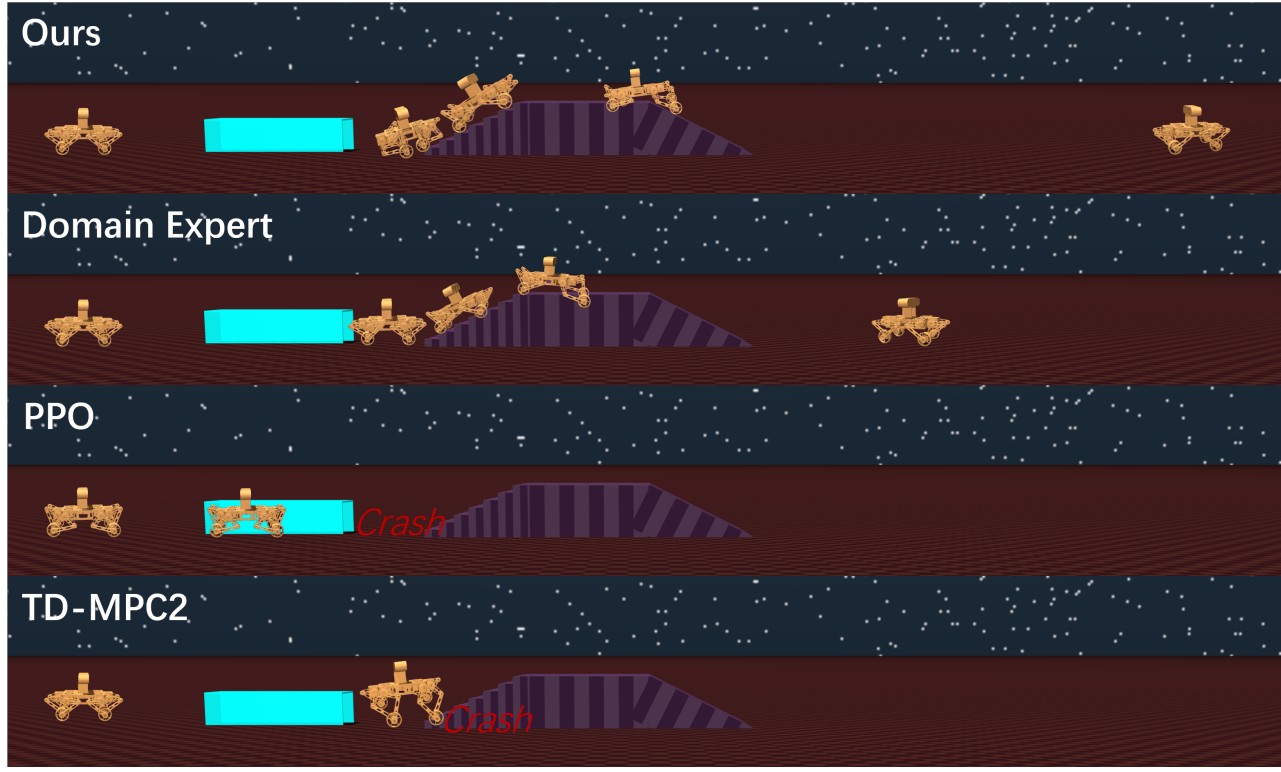

Figure 12. Visualization of comparison with single-robot baselines of Wheel-leg experiment, containing four frames in $T = 0, 2, 4, 6, 8$ s respectively. "Crash" means the algorithm makes the robot crash at this point.

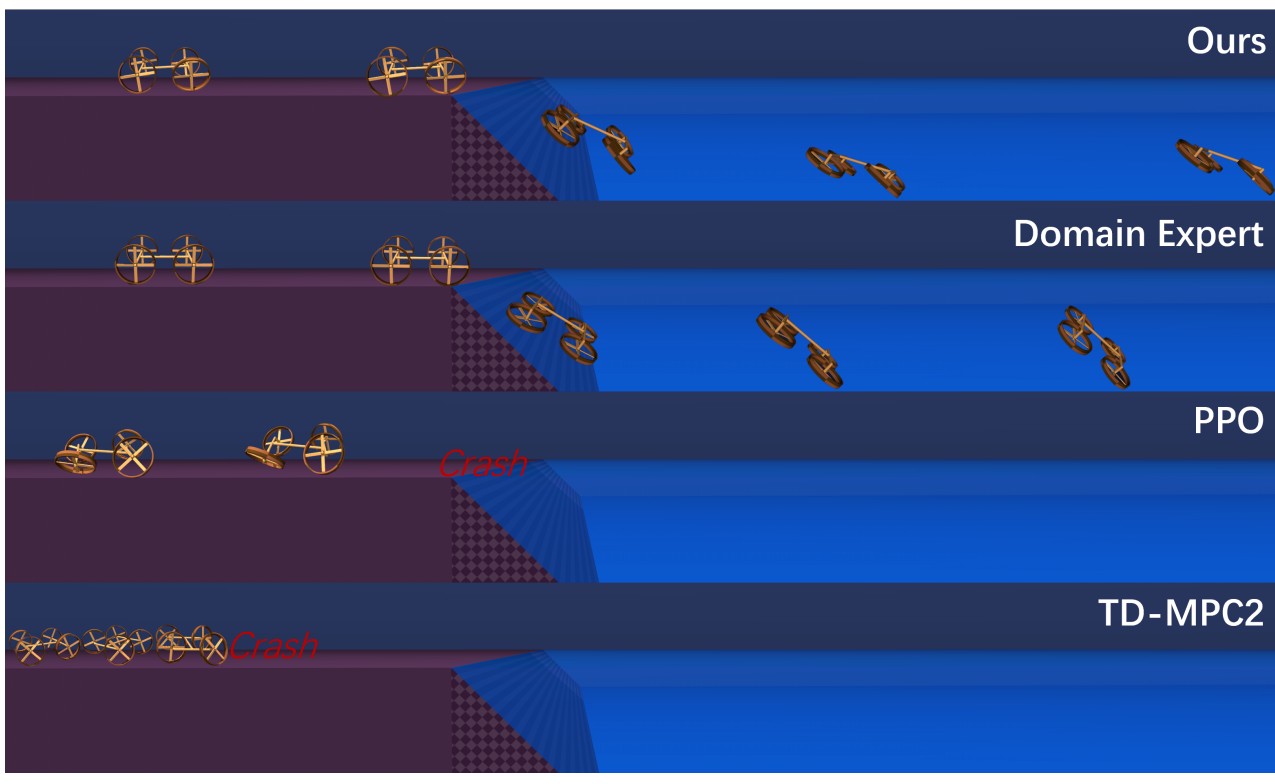

Figure 13. Visualization of comparison with single-robot baselines of Wheel-fly (swim) experiment, containing four frames in $T = 8, 11, 14, 17, 20$ s respectively. "Crash" means the algorithm makes the robot crash at this point.

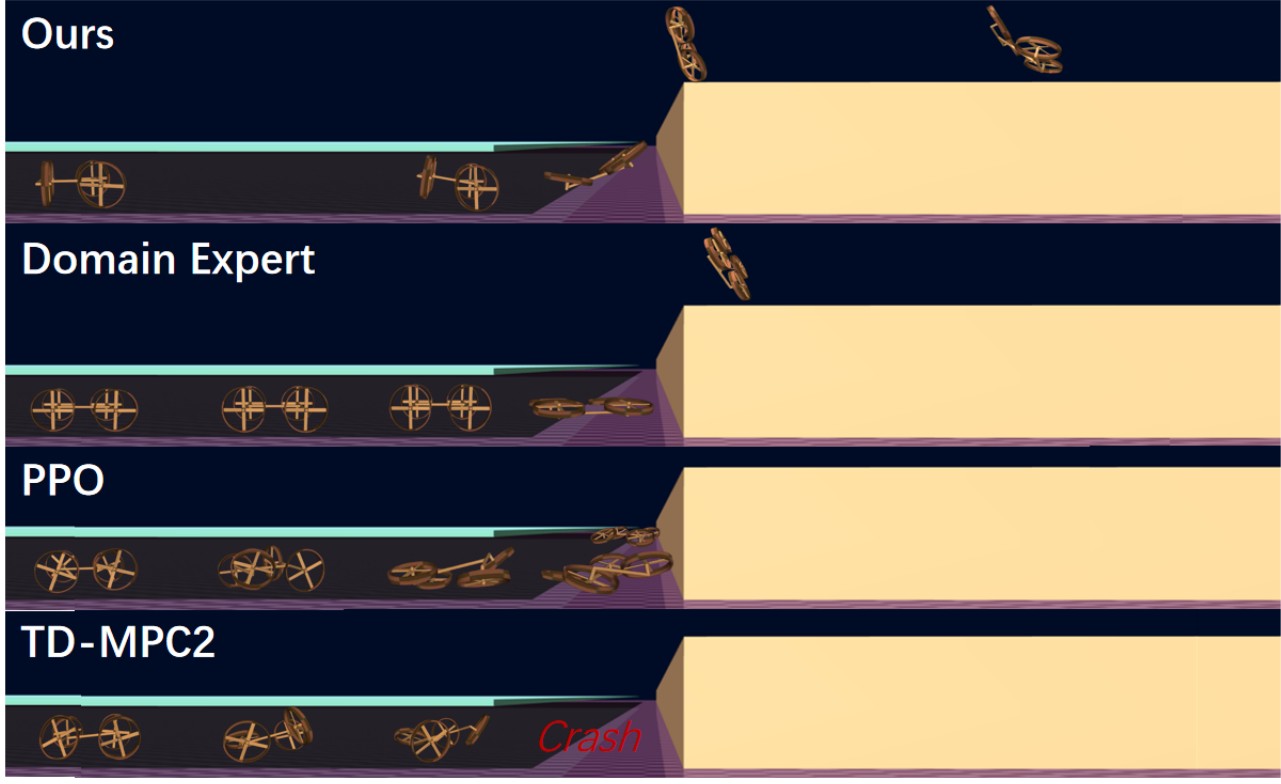

Figure 14. Visualization of comparison with single-robot baselines of Wheel-fly (Mount) experiment, containing four frames in $T$=0, 3, 6, 9, 12 s respectively. "Crash" means the algorithm makes the robot crash at this point.

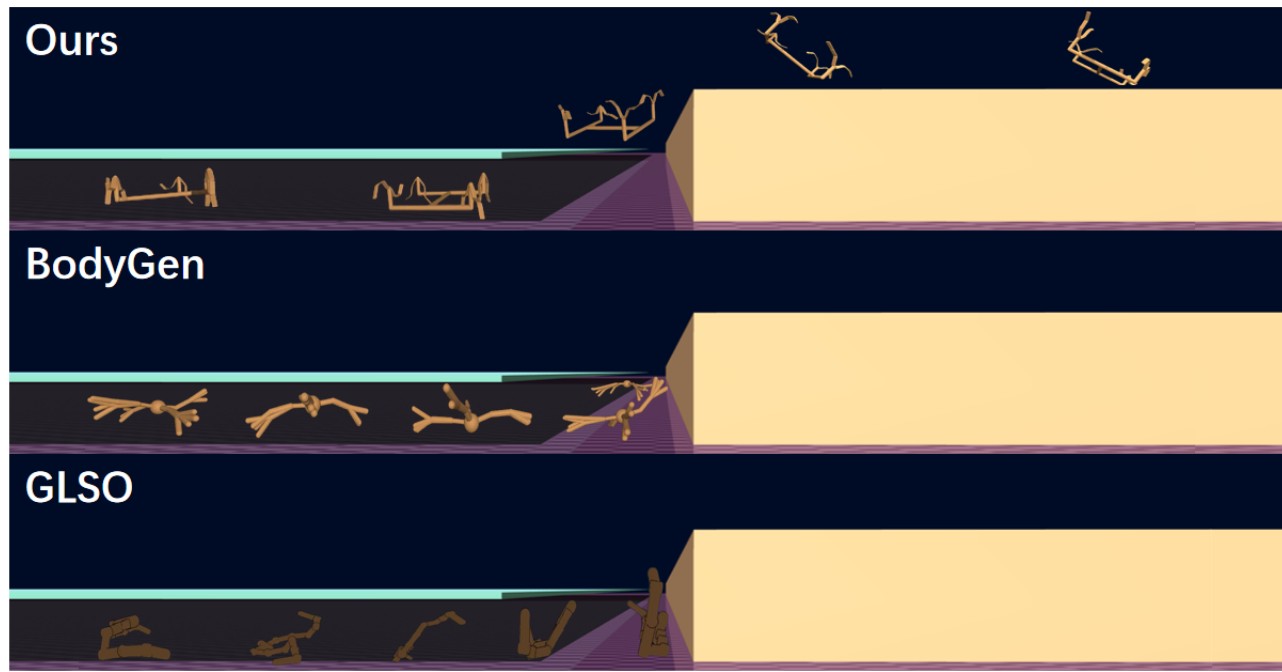

*Figure 15.* Visualization of comparison with morphology-co-design baselines of Fly-walk (Mount) experiment, containing four frames in $T = 0, 5, 10, 15, 20$ s respectively.

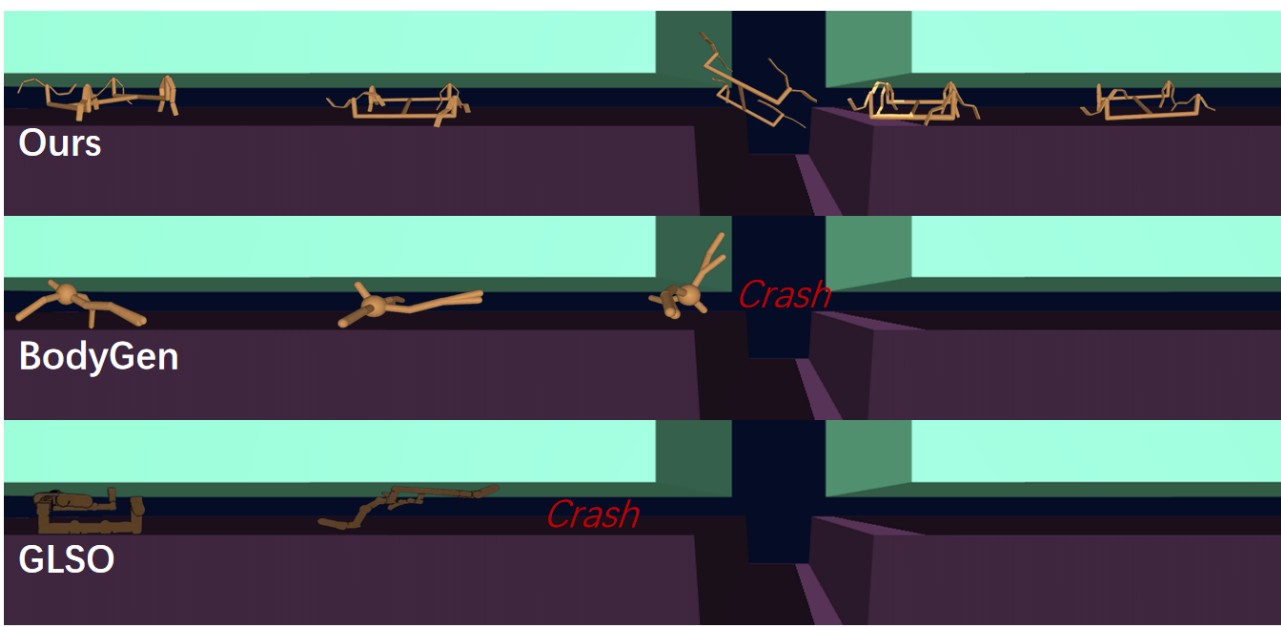

*Figure 16.* Visualization of comparison with morphology-co-design baselines of Fly-walk (Pit) experiment, containing four frames in $T = 0, 5, 10, 15, 20$ s respectively. "Crash" means the algorithm makes the robot crash at this point.

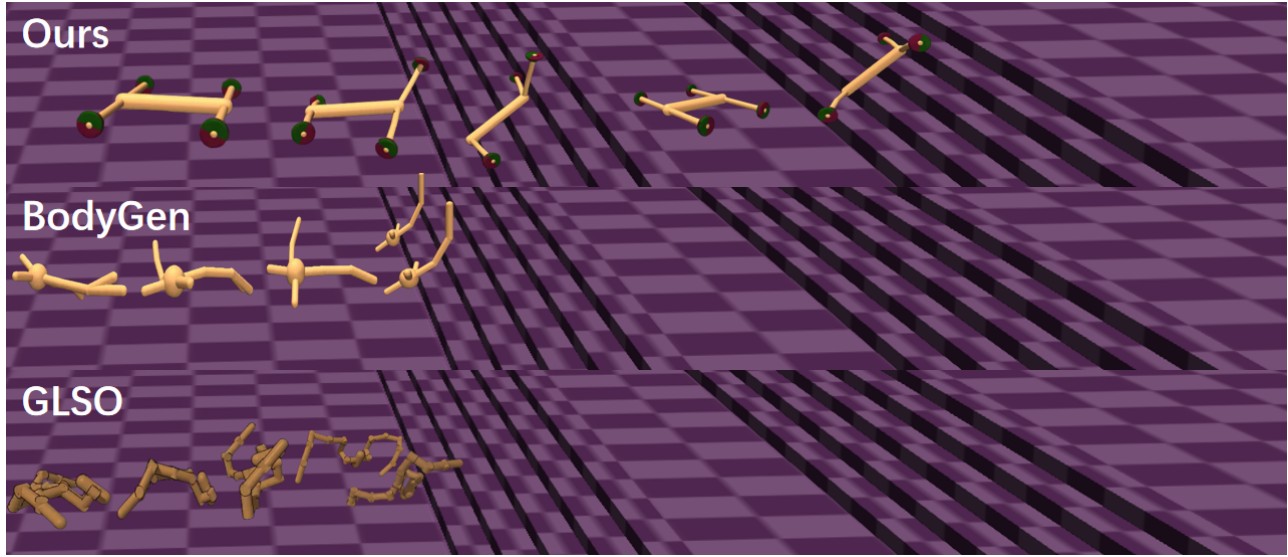

*Figure 17.* Visualization of comparison with morphology-co-design baselines of Car-leg (Stairs) experiment, containing four frames in $T = 0, 7, 14, 21, 28$ s respectively.

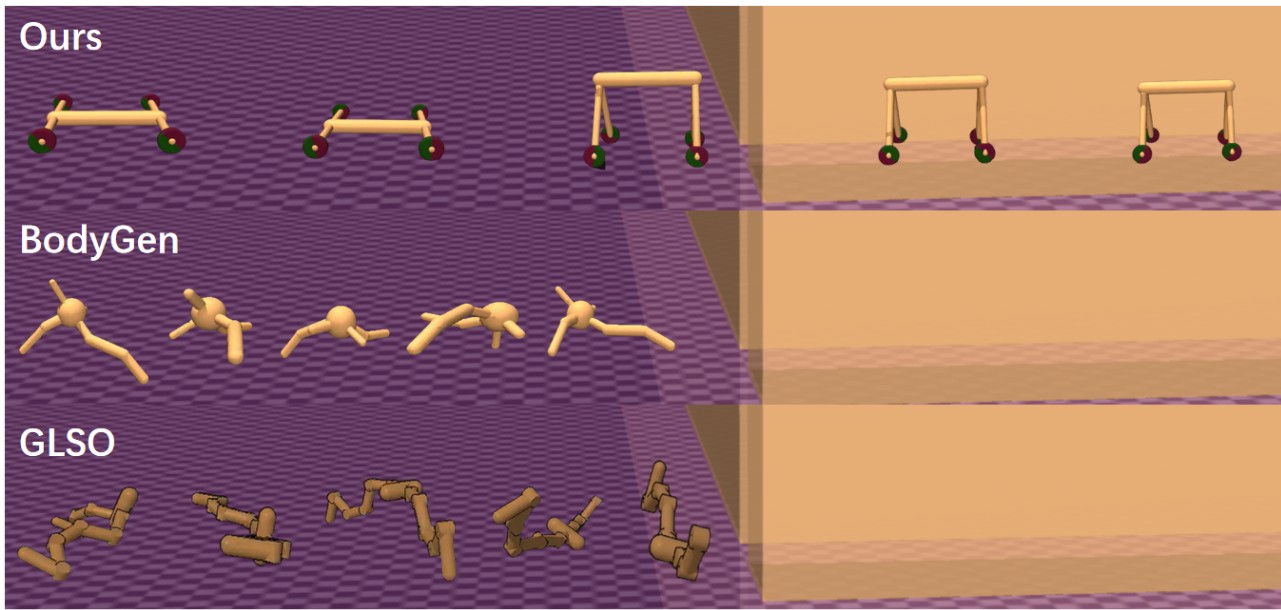

*Figure 18.* Visualization of comparison with morphology-co-design baselines of Car-leg (Narrow) experiment, containing four frames in $T = 0, 5, 10, 15, 20$ s respectively.

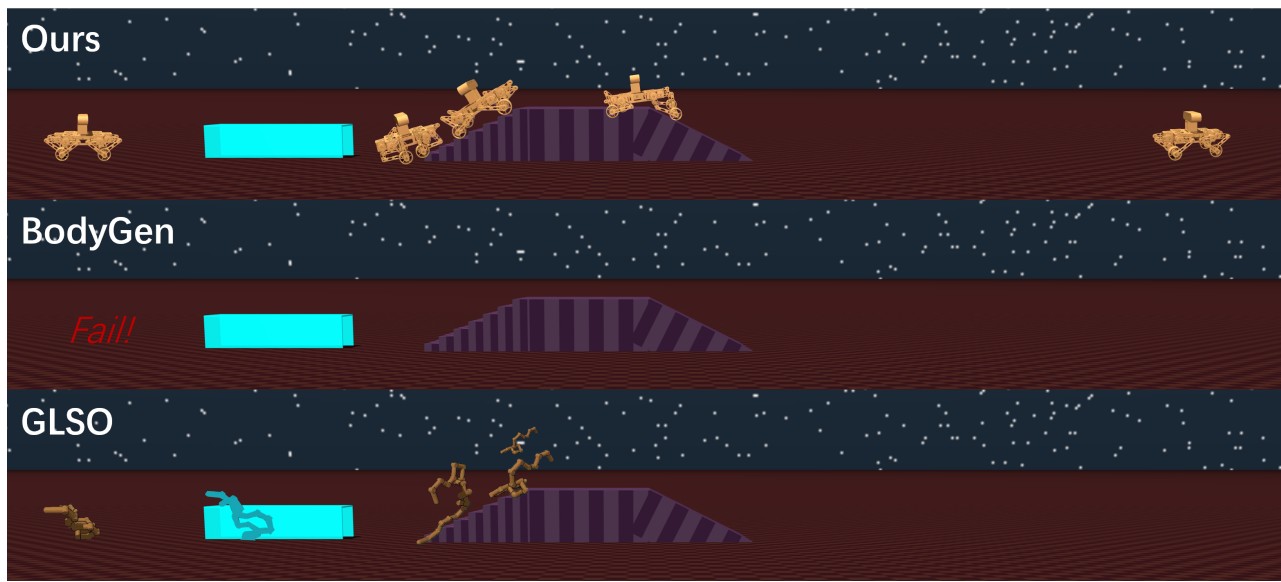

*Figure 19.* Visualization of comparison with morphology-co-design baselines of Wheel-leg experiment, containing four frames in $T = 0$, 2, 4, 6, 8 s respectively. "Fail" means the algorithm fails to output effective results in this experiment.

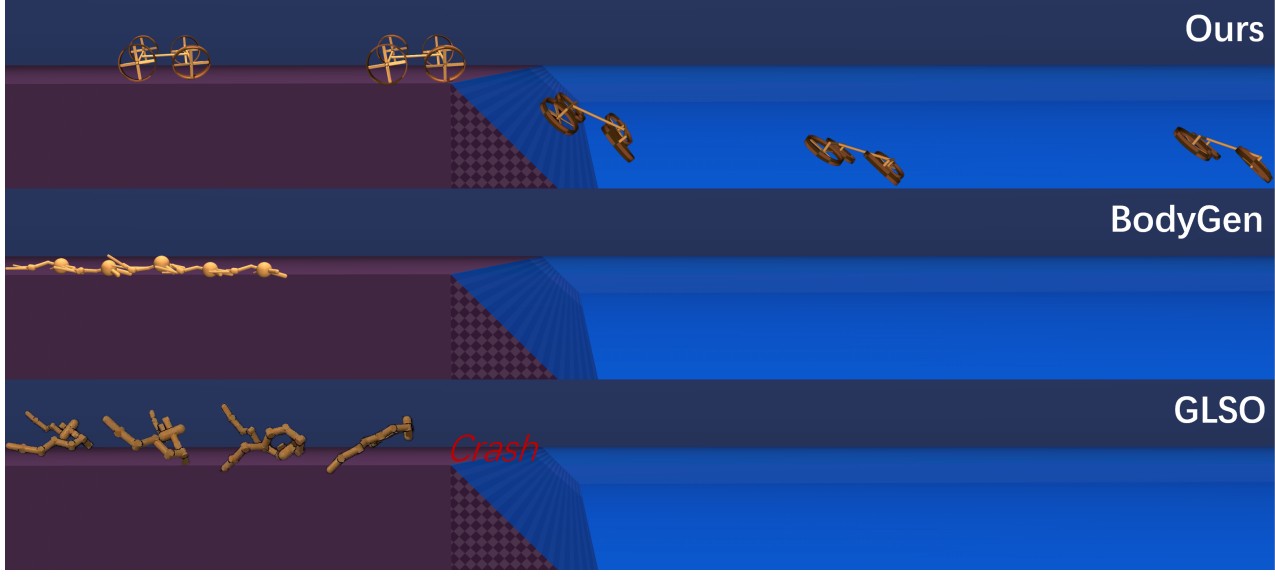

*Figure 20.* Visualization of comparison with morphology-co-design baselines of Wheel-fly (swim) experiment, containing four frames in $T = 8$, 11, 14, 17, 20 s respectively. "Crash" means the algorithm makes the robot crash at this point.

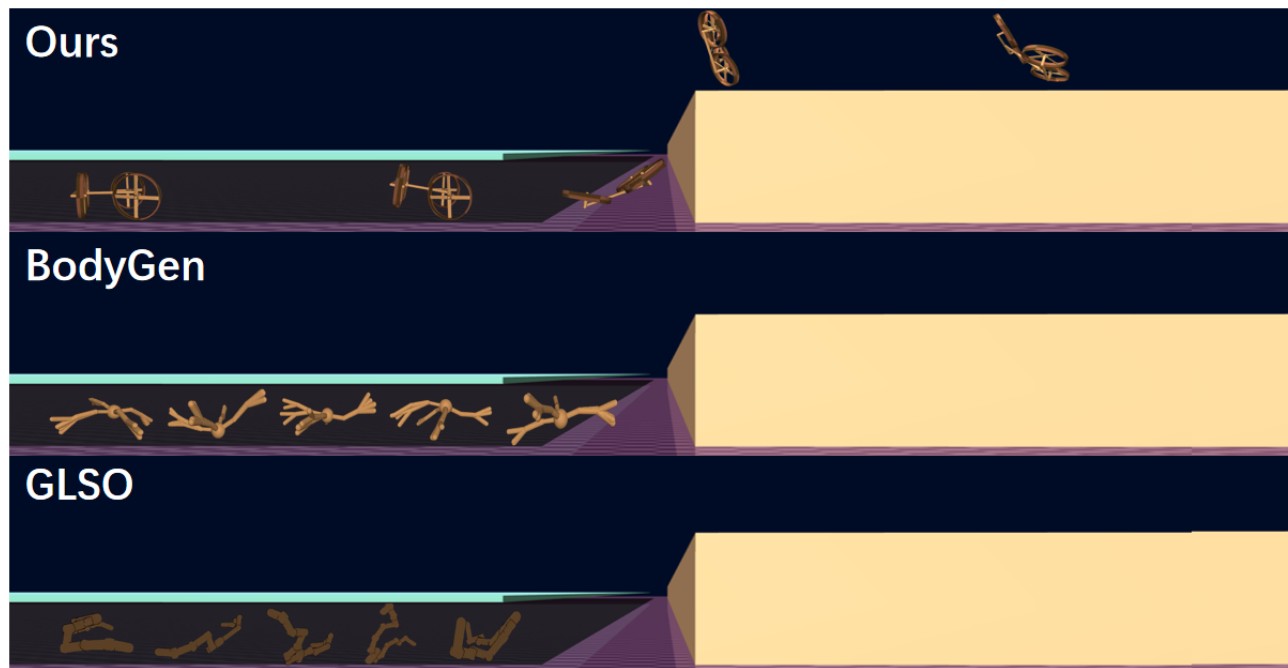

*Figure 21.* Visualization of comparison with morphology-co-design baselines of Wheel-fly (Mount) experiment, containing four frames in $T = 0, 3, 6, 9, 12$ s respectively.

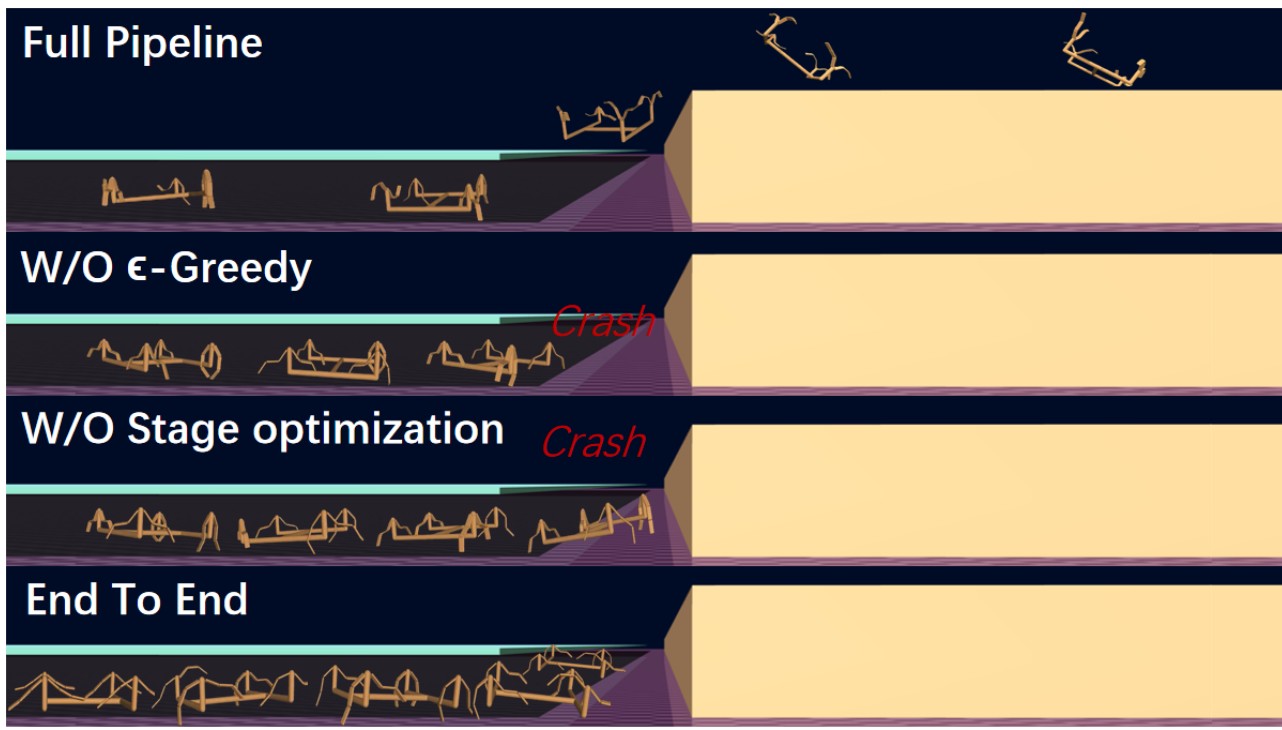

*Figure 22.* Visualization of comparison with ablation studies of Fly-walk (Mount) experiment, containing four frames in $T = 0, 5, 10, 15, 20$ s respectively. "Crash" means the algorithm makes the robot crash at this point.

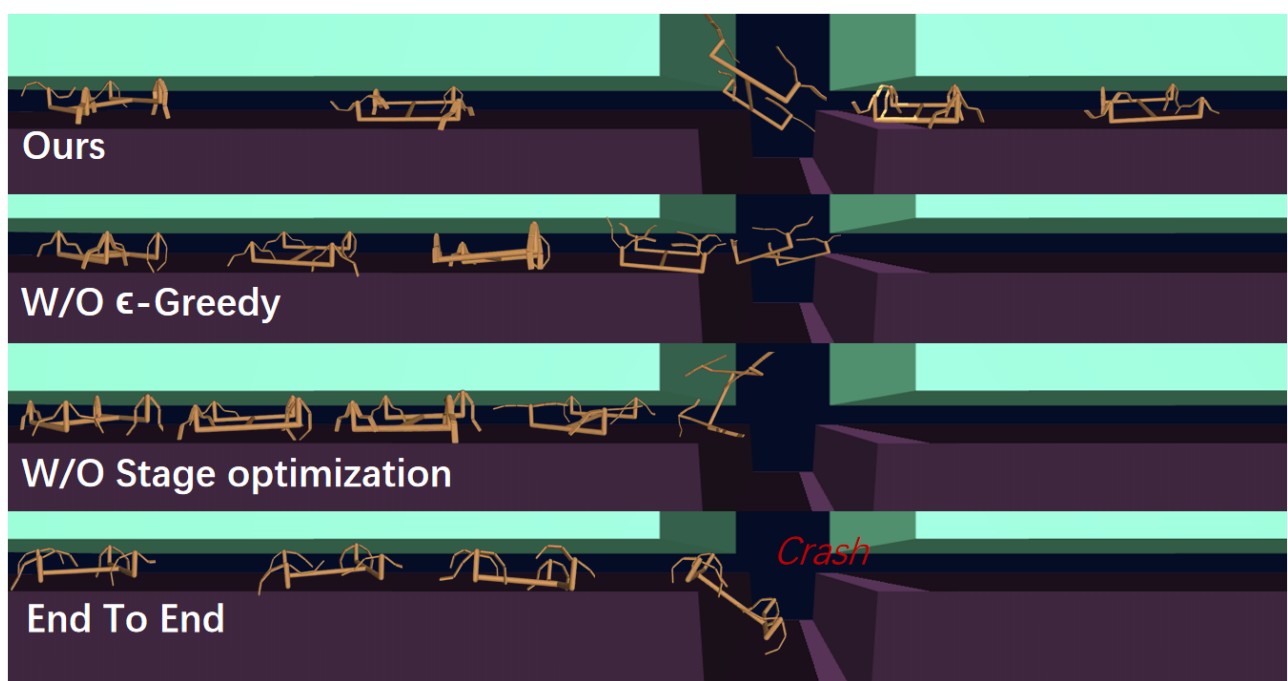

*Figure 23.* Visualization of comparison with ablation studies of Fly-walk (Pit) experiment, containing four frames in $T = 0, 5, 10, 15, 20$ s respectively. "Crash" means the algorithm makes the robot crash at this point.

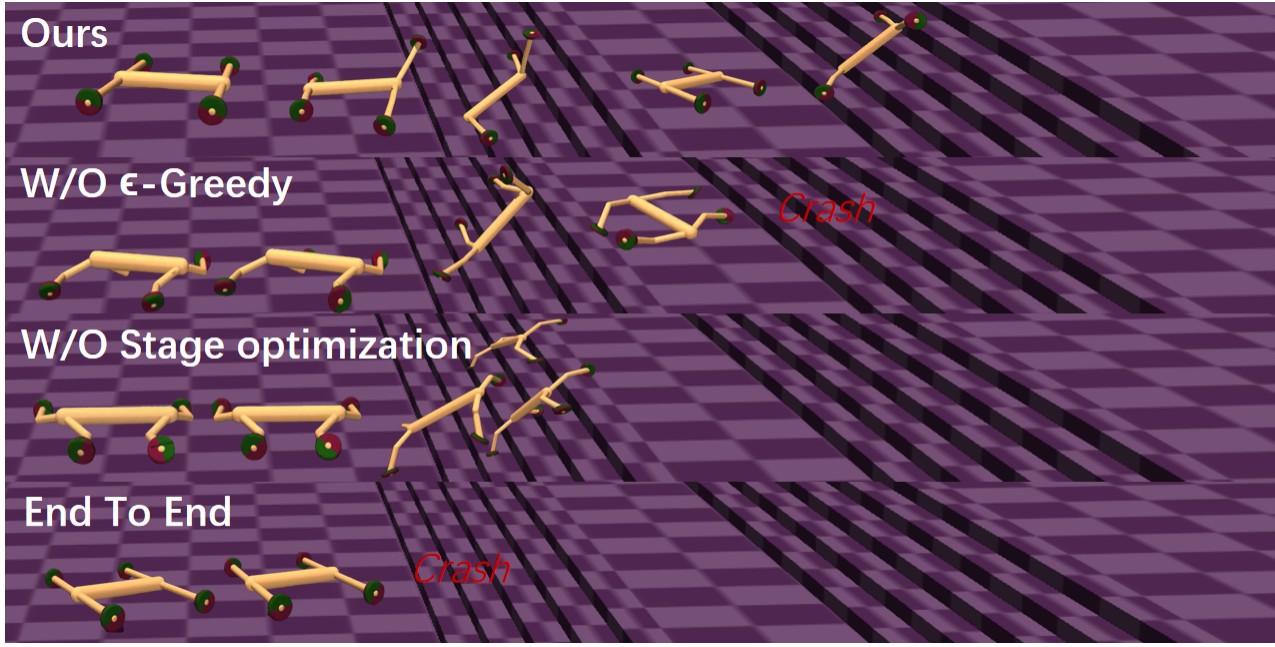

*Figure 24.* Visualization of comparison with ablation studies of Car-leg (Stairs) experiment, containing four frames in $T = 0, 7, 14, 21, 28$ s respectively. "Crash" means the algorithm makes the robot crash at this point.

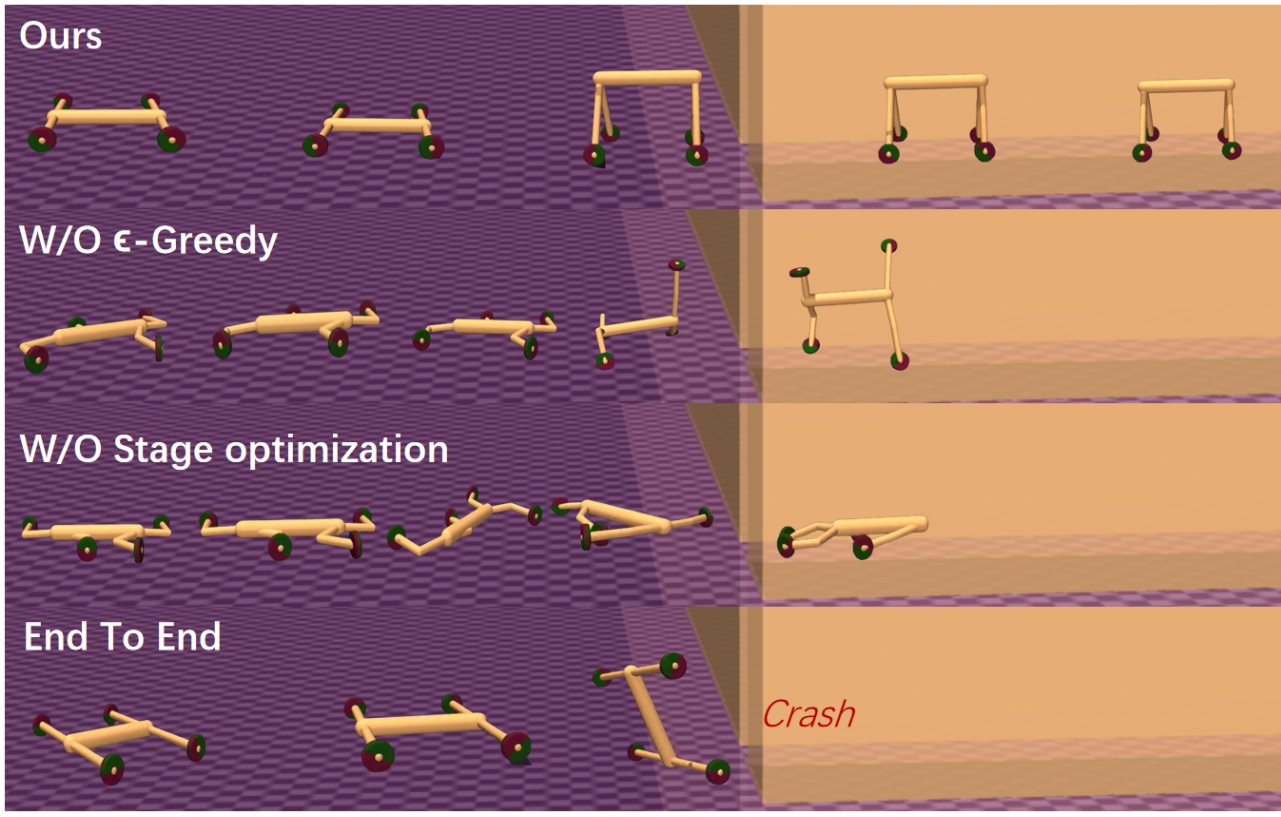

*Figure 25.* Visualization of comparison with ablation studies of Car-leg (Narrow) experiment, containing four frames in $T = 0, 5, 10, 15, 20$ s respectively. "Crash" means the algorithm makes the robot crash at this point.

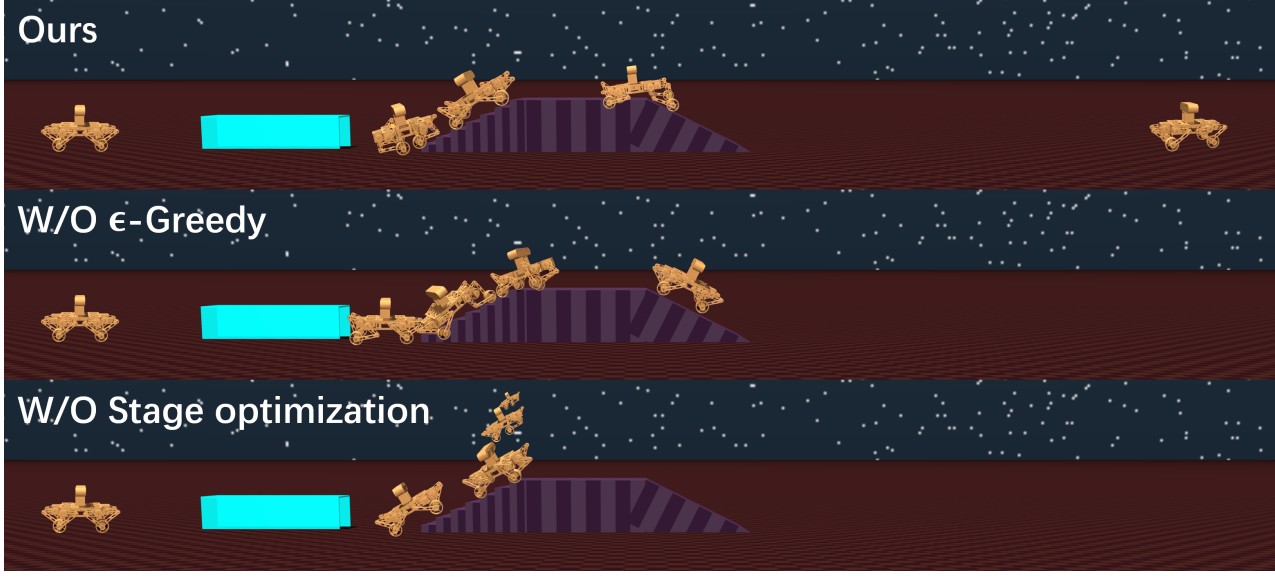

*Figure 26.* Visualization of comparison with ablation studies of Wheel-leg experiment, containing four frames in $T = 0, 2, 4, 6, 8$ s respectively.

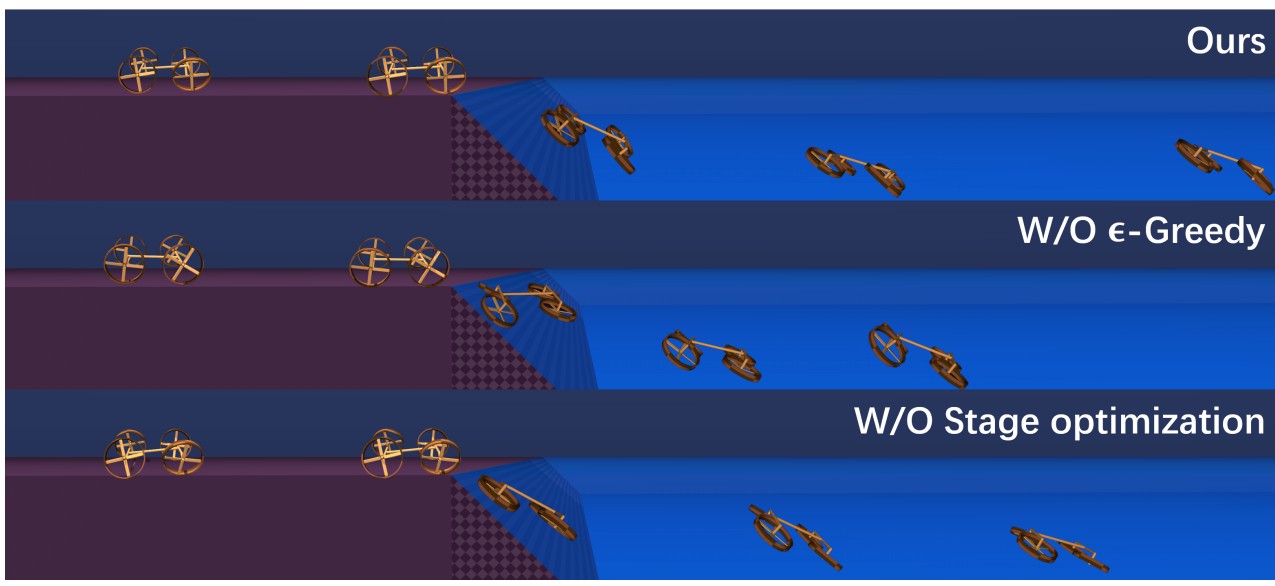

*Figure 27.* Visualization of comparison with ablation studies of Wheel-fly (swim) experiment, containing four frames in $T = 8, 11, 14, 17, 20$ s respectively.

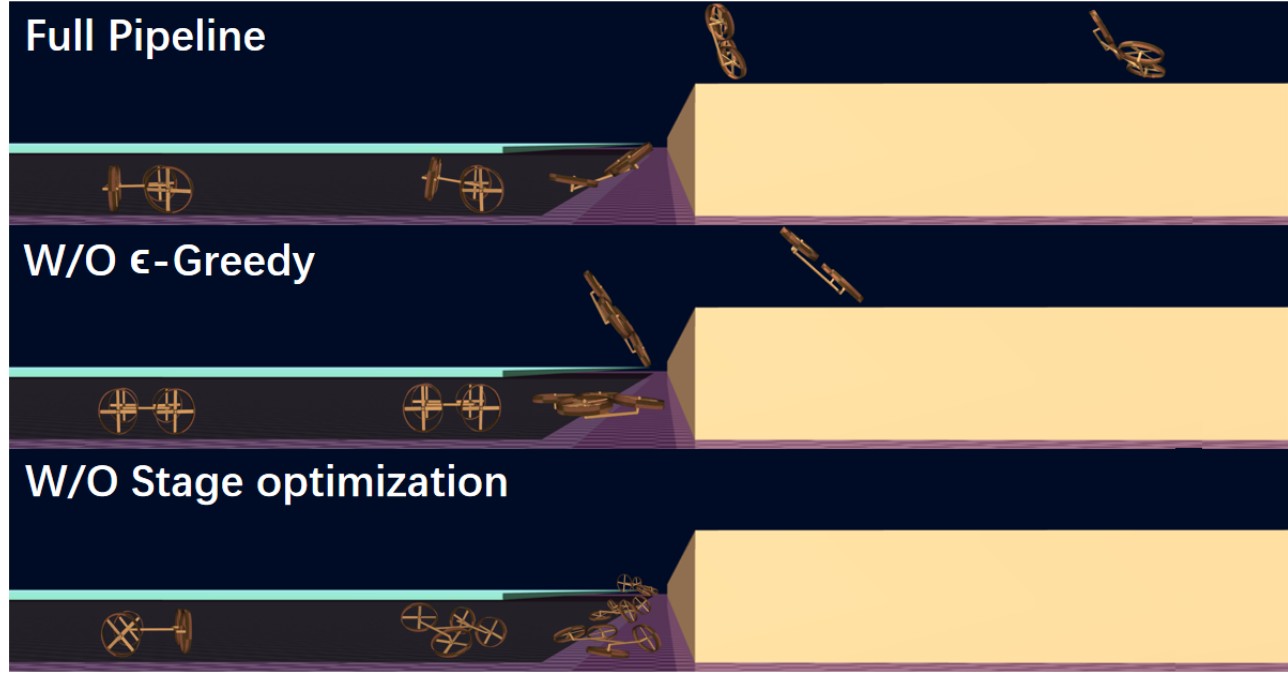

*Figure 28.* Visualization of comparison with ablation studies of Wheel-fly (Mount) experiment, containing four frames in $T = 0, 3, 6, 9, 12$ s respectively.

