# OpenReview forum: "Learning to Reconfigure: Configuration-Control Co-optimization of Reconfigurable Robots for Heterogeneous Locomotion"
_ICML.cc/2026/Conference — ICML 2026 regular_

### Official Review · Reviewer_2v1m · 2026-02-27

**Soundness:** 3
**Presentation:** 3
**Significance:** 3
**Originality:** 3
**Overall Recommendation:** 5
**Confidence:** 4

**Summary:**

This paper explores co-designing reconfigurable robots for heterogeneous environments. Whereas previous works either focused on single-robot design which cannot adapt to a dynamic environment or reconfigurable designs that rely on human-engineered configurations, this paper bridges the two and develops a two part approach. First, the authors propose a method to optimise specialised primitives for sub-terrains using a multi-tail architecture. Then, they propose a scheduler that learns to switch configurations based on global task progress. The authors compare their method against two types of baselines in a variety of locomotion tasks that require switching between walking (different terrains), flying and swimming. They find that their method outperforms these baselines significantly.

**Compliance With Llm Reviewing Policy:**

Affirmed.

**Final Justification:**

I thank the authors for their engagement in the rebuttal period. I believe they have mostly addressed my concerns and therefore raised my score accordingly.

**Key Questions For Authors:**

•	If possible, the authors should consider implementing a PPO baseline with a fairer training regime to demonstrate that reconfigurable morphologies are essential for heterogeneous environments.
•	If possible, the authors should consider replicating their experiments across several seeds and reporting statistical significance of their results.
•	It would be helpful if the authors could clarify some details about their method which are a bit unclear right now (listed above).
•	It would be helpful if the authors could comments on the different base morphologies of the robots in Figure 4.
•	It would be helpful if the authors could comment on why figure 5 only display ablation results for 4 tasks and not all of them?

**Limitations:**

yes

**Strengths And Weaknesses:**

Strengths:
- Soundness: The authors introduce very interesting and suitable environments to test their method on, making the experiments well-designed to test the key motivations of the paper. The baselines are mostly very well-suited and the authors include an honest discussion of limitations in their conclusion. The explanation behind the method is well articulated and the ablation studies justify key design choices.
- Presentation: The presentation of the paper is very clear, well-written and easy to follow. It is positioned clearly in the context of prior/concurrent literature.
- Significance: The problem of adaptive and optimised morphology for heterogeneous environments is a very important and relevant problem. Considering both co-design of morphology and reconfigurability simultaneous presents a genuinely new and exciting direction.
- Originality: The work presents new insights and highlights the limitations of existing methods of both co-design of morphology and reconfiguration methods. The contributions are clearly distinguished from existing literature and the novelty is well justified.

Weaknesses:
- Soundness (baselines): While the authors note that exact baselines are difficult, I think that more effort should have been made to make the comparison to PPO fairer. In particular, the method proposed in the paper gets equal training on each subterrain so it is unsurprising that it has a much better success rate. It would be much fairer to compare the approach to either a) PPO which has equal traning on each subterrain (e.g. switching starting positions at each episode) or b) a skill-conditioned RL baseline using the “global task information” that the proposed approach receives (or both)! The key claim of the paper is that we need robots to adapt configurations to different terrains but if PPO could solve this with the same training paradigm as the main method and without the overhead cost of a scheduler network to switch between morphologies this would significantly impact the claim the authors are making.
- Soundness (baselines): The authors omit details about the steps they took to ensure a fair experimental setup for all baselines. For instance, did you ensure all baselines receive the same budget? Do they all have the same observations? E.g. the scheduler network receives “global task information”. It’s unclear what this is, whether it is privileged information and whether any of the other baselines have access to it? Similar questions for other training hyperparameters? I’m sure that the authors did take measures for this but they omit too many details.
- Presentation (clarity): The method was very well written but some details were slightly unclear. Firstly: how is h_shared learned? Is it just a reconstruction loss? Secondly, in stage 1 when only the sub-task observation is input to the extractor, is the rest of the state just masked? Finally, I find it unclear at what timesteps Stage 1 and Stage 2 are happening at? It’s clear that the rewards for the actor-critic transitions are counted at the episode level but does that mean that the lock mask and angles are only changed at the beginning of each episode or is that all timesteps are assigned the episode-level reward? It’d be helpful to add the episode length into the hyperparameters and total evaluation budget to help understand this too.  I’d appreciate some clarification on these.
- Soundness (baselines): It would be nice to have more intuition of how the authors picked the domain-expert configurations. It could have been fairer to test their method against the same number of K domain-expert primitives or even K random primitives. It would be interesting to combine these with the scheduler network to decouple the performance of their co-design and global control.
- Soundness (technical soundness): The experiments are only presented for one seed. The authors should run the experiments for several seeds to test the significance of their results.
- Soundness (experiments): Why does figure 5 only display ablation results for 4 tasks and not all of them?
- Soundness (limitations): While the authors acknowledge several limitations in their conclusion, they do not discuss that they assume global knowledge of the task by being able to reset the robot to the same initial state in each task and thus be able to easily find the k primitives. It’s unclear how well this method would generalise to a more difficult and varied suite of tasks with potentially many more subterrains.
- Presentation (clarity): In Figure 4 it looks like the base morphology for each of the baselines. Why is that?
- Presentation (clarity): In Section 6.4, it wasn’t clear what the “staged optimisation mechanism” was at first, it might be helpful to refer to this earlier in the methods section or explain it a bit again. Figure 5 could be combined into table 3 if needed to make space.

---

> ### Author Rebuttal · Authors · 2026-03-30
>
> **1. (W1 & Q1) Fair comparison with PPO baselines**
>
> Thank the reviewer for raising this point. We would like to clarify that our PPO and TD-MPC2 baselines already implement the training paradigm the reviewer suggested in (a). When training these baselines, we do not solely spawn the robot at the beginning of the map. Instead, we explicitly initialize the robot across different sub-terrains with equal sampling steps, ensuring the policy experiences all physical environments equally.
>
> However, due to the conflicting gradients in heterogeneous terrains, a single configuration cannot reconcile these physically incompatible requirements, leading to a local optimum which only completes the easiest sub-task. This failure under equal training validates our claim: reconfiguration is a physical prerequisite for multi-physics tasks, not something that can be overcome by better data sampling.
>
> Regarding the "global task information", we provide a detailed explanation in "Alignment of baselines" below.
>
> **2. (W2) Alignment of baselines**
>
> - **Equal Training Budgets**: We ensured that our method and all baselines (including PPO and TD-MPC2) received the same training budget. Specifically, every method was allocated an ample and equal number of environment sampling steps to ensure they all reached full convergence before their final behaviors were evaluated and compared.
> - **No Privileged Information**: To clarify, "global task information" refers to the body-centric terrain height field perceived by the robot as it moves. We initially used the word "global" because the task requires traversing the global terrain map. This same height field observation is provided as input to all baselines, and there is no privileged information given to our method. To eliminate ambiguity, we will change the word to a more precise term.
>
> **3. (W3 & Q3) Details of implementation**
>
> To clarify your questions directly:
> - Vector $h_{shared}$: $h_{shared}$ is not learned via an auxiliary reconstruction loss. It is simply the intermediate latent feature output by the shared feature extractor.
> - Input Masking in Stages 1 & 2: The reviewer's interpretation is correct. When specific inputs are not provided during Stage 1 and Stage 2, the shared extractor masks these missing components using zero-padding.
> - Timing of Stages 1 & 2: Stages 1 and 2 occur once in each episode. The lock mask and locking angles are decided at this initial step and remain fixed throughout the rest of the episode.
>
> **4. (W4) Design of domain-expert baselines**
>
> Thank the reviewer for raising this question. Since this is a common question among reviewers, please refer to answer 2 to reviewer kS5a.
>
> **5. (W5 & Q2) Variance on random seeds**
>
> Thank the reviewer for raising this question. Since this is a common question among reviewers, please refer to answer 3 to reviewer kS5a.
>
> **6. (W6 & Q5) More tasks in end-to-end ablation study**
>
> Based on the suggestion from the reviewer, we report the statistics for the remaining tasks in the table below:
> |Method|Wheel-leg|Wheel-propeller (mount)|Wheel-propeller (swim)|
> |-|-|-|-|
> |Ours|**9.148**|**7.368**|**3.967**|
> |End-to-end|0.225|0.625|0.324|
>
> As shown, our method significantly outperforms the end-to-end method in these tasks, which corresponds to the trends of the four tasks reported in our paper.
>
> **7. (W7) Initial state setup**
>
> Thank the reviewer for raising this point. We would like to clarify that we do not reset the robot to the same initial state, nor do we assume static global knowledge. Instead, our training process heavily randomizes both the robot's initial states (e.g., positions and yaw directions) and the geometric parameters of the terrains (e.g., mountain heights and pit width). This domain randomization ensures that our method does not overfit to a specific trajectory, but instead learns robust strategies capable of handling various environments.
>
> **8. (W8 & Q4) Different morphologies in Fig.4**
>
> Thank the reviewer for this observation. First, we conducted a new comparison providing these co-design baselines with the same initial morphology as our robot. They still failed to match our performance (please refer to answer 1 to reviewer a52R for detailed results).
>
> Second, regarding the original visualizations in Fig. 4, the reason the base morphologies differ is that these baseline methods are co-design algorithms which optimize both the robot's morphology and its control policy from scratch. We visualized their final converged shapes to demonstrate that current SOTA reconfiguration-free co-design methods struggle to find a viable unified body plan for heterogeneous tasks, often converging to sub-optimal or compromised designs.
>
> **9. (W9) Suggestion on writing**
>
> We appreciate these helpful suggestions and will update the manuscript accordingly.

---

> > ### Author Rebuttal · Reviewer_2v1m · 2026-04-01
> >
> > Thank you to the authors for clarifying the details of this paper, I am satisfied that my concerns have been adequately addressed and have adjusted my score accordingly. I'd like to encourage the authors to revise the manuscript to add clarifications based on the questions above.

---

> > > ### Author Response · Authors · 2026-04-01
> > >
> > > Thank the reviewer for the valuable feedback. We will update our manuscript according to the reviewer’s suggestions.

---

### Official Review · Reviewer_a52R · 2026-03-05

**Soundness:** 3
**Presentation:** 4
**Significance:** 3
**Originality:** 3
**Overall Recommendation:** 4
**Confidence:** 3

**Summary:**

This paper addresses the problem of automatically discovering joint-locking configurations and control policies for reconfigurable robots operating in heterogeneous locomotion tasks that span terrestrial, aerial, and aquatic domains. The authors propose a hierarchical pipeline called "Learning to Reconfigure," consisting of two stages. In the first stage, multiple low-level primitives are trained independently on different sub-tasks, each using a multi-tail policy architecture that separates the optimization of structural decisions (which joints to lock and at what angles) from continuous motor control. In the second stage, a high-level scheduler is trained via PPO to dynamically switch among the frozen primitives based on global task progress. To support training across diverse physical media, the authors build a multi-physics simulation environment on top of MuJoCo with simplified fluid dynamics for aerial and aquatic interactions. Experiments on seven heterogeneous locomotion tasks (e.g., fly-walk, car-leg, wheel-fly) demonstrate substantial performance gains over single-robot control baselines (PPO, TD-MPC2, domain-expert configurations) and morphology co-design methods (BodyGen, GLSO), measured by normalized traversal progress. Ablation studies validate the contributions of the hierarchical decomposition, epsilon-greedy exploration, and staged optimization mechanisms.

**Compliance With Llm Reviewing Policy:**

Affirmed.

**Final Justification:**

My major concerns are resolved by the additional comparisons with BodyGen and GLSO, as well as the further explanations made by the authors. I therefore decided to raise my score to 4.

**Key Questions For Authors:**

How are sub-tasks defined and how is the number of primitives K determined? Specifically, is the decomposition based merely on the physical medium (land/air/water)? If so, this coarse-grained decomposition by medium type may not scale to more complex terrains where multiple sub-terrains share the same medium (land) but demand very different locomotion strategies. Clarification on whether an automatic decomposition mechanism exists or is feasible would strengthen the paper.

**Limitations:**

yes

**Strengths And Weaknesses:**

**Strengths:**
1. **Novel problem formulation with a well-motivated hierarchical design.**
To the best of my knowledge, the automated discovery of reconfiguration strategies is a relatively unexplored problem. The proposed two-stage hierarchy—first training specialized primitives per sub-task, then learning a scheduler to switch among them—is an intuitive and effective decomposition. This decoupling avoids the gradient conflict between long-horizon reconfiguration planning and high-frequency motor control, which is empirically validated by the significant performance gap over the end-to-end baseline in the ablation study (Fig. 5).
2. **Comprehensive experiments.**
The evaluation spans seven heterogeneous locomotion tasks across terrestrial, aerial, and aquatic physical domains, providing broad coverage of the problem space. Two categories of baselines are compared: single-robot control algorithms (PPO, TD-MPC2, domain-expert) and morphology co-design methods (BodyGen, GLSO)—though the fairness of the latter comparison is debatable, as discussed below. The reported improvements (5.95x and 9.99x average progress) are substantial.
3. **Rich qualitative analysis with non-trivial design insights.**
The appendix provides extensive visualizations (Fig. 6–28), including task panoramas, discovered configuration galleries, and frame-by-frame comparisons across all methods, which greatly aid in understanding the method's behavior. Notably, the paper reveals counter-intuitive yet effective configurations—such as the robot flipping its rear wheels forward for tunnel propulsion in Wheel-fly (Mount)—demonstrating that the automated pipeline can explore regions of the design space that human experts would not typically consider.

**Weaknesses:**
1. **Unfair comparison with co-design baselines.**
The comparison with BodyGen and GLSO (Tab. 2) is fundamentally asymmetric. From the text and appendix visualizations, these baselines appear to design morphologies from scratch without access to the pre-defined embodiment, while the proposed method starts from a human-designed robot already capable of multi-modal locomotion (e.g., wheels + propellers). Moreover, baselines output a single static morphology whereas the proposed method deploys multiple switchable configurations. The 9.99x gap thus conflates the advantage of a capable starting embodiment, the advantage of reconfiguration, and potential search space limitations of the baselines. A fairer comparison would adapt co-design baselines to the same setting—e.g., given the same base robot, optimize a single fixed joint-locking configuration—or run them per sub-task and integrate with the proposed scheduler. If the authors could demonstrate that, under a fair comparison where existing co-design methods are adapted to the same problem setting, the proposed method still achieves superior performance, I would be willing to raise my score.
2. **"Co-design" terminology is somewhat misleading.**
"Co-design" conventionally refers to joint optimization of robot morphology (topology, geometry) and control. This paper takes a fixed embodiment and only optimizes joint-locking patterns and control—a substantially smaller search space. Using "co-design" throughout, especially when directly comparing against true co-design methods, overstates the scope. More precise terminology such as "configuration-control co-optimization" would be more appropriate.
3. **Missing details on variable inputs across stages.**
The three inference stages of each primitive have progressively expanding inputs, yet the paper describes a "shared feature extractor" without specifying how variable-dimensionality inputs are handled (zero-padding, selective branches, or stage-specific interfaces). These details are essential for reproducibility.

---

> ### Author Rebuttal · Authors · 2026-03-30
>
> **1. (W1) Initial guess of co-design baselines**
>
> We thank reviewer kS5a, a52R, and 2v1m for raising this question.
>
> **Same initial guess of co-design baselines**: To ensure a fair comparison, we conducted an additional experiment as the reviewers suggested. In more detail, we evaluated both BodyGen and GLSO by providing them with the exact same robot morphology as their initial guess. The results are exhibited in the following table:
> |Method|Fly-walk (mount)|Fly-walk (pit)|Car-leg (stairs)|Car-leg (narrow)|Wheel-leg|Wheel-propeller (mount)|Wheel-propeller (swim)|
> |-|-|-|-|-|-|-|-|
> |Ours|**1.978**|**1.417**|**0.872**|**2.618**|**9.148**|**7.368**|**3.967**|
> |BodyGen|0.452|0.443|0.511|0.484|Fail|0.515|0.293|
> |GLSO|0.496|0.483|0.367|0.535|0.264|0.502|0.355|
>
> From the results, we can observe that even with this starting advantage, the baselines still fail to utilize the morphological potential, and do not perform significantly better than training from pure scratch. Further analysis (combined with the ablation studies on end-to-end method) exhibits that they suffer from severe conflicting gradients inherent in long-horizon heterogeneous tasks. Without a mechanism to navigate this non-convex optimization space, they quickly degrade to trivial solutions capable of traversing only the easiest terrain segments. This confirms that our substantial performance gap stems from our algorithmic design, not the starting embodiment.
>
> **Joint-locking concerns**: Regarding the suggestion on joint-locking, we would like to note that from a control perspective, joint-locking can be functionally achieved by feeding a constant target to a position servo (which is our implementation). Therefore, the controllers in the baseline methods theoretically already possess the capacity to learn a "joint-fix" behavior. Our method's unique advantage lies not simply in possessing this ability, but rather in our architectural decoupling. We explicitly separate the low-frequency "structural lock mode" (lock mask & angle) from the high-frequency "dynamic control mode" (control signal). Baselines attempt to learn both within a single, entangled policy, leading to severe optimization difficulties (Tab. 1 & 2). Our decoupled architecture simplifies the search space, enabling the highly effective joint optimization that baselines struggle to achieve.
>
> **2. (W2) Co-design Terminology**
>
> Thank the reviewer for this suggestion. It is correct that our pipeline optimizes the configuration (logical topology of the robot), not the physical morphology. We will replace "co-design" with "configuration-control co-optimization" as suggested.
>
> **3. (W3) Details on variable inputs across stages**
>
> To handle the inputs across the three stages, our shared feature extractor utilizes a linear dimension expansion combined with zero-padding. Specifically, the extractor first expands the dimension of sub-task observation ($o_t$) and design observation ($o_d$), and then concatenates them with the state observation ($o_s$). During Stage 1 (where $o_d$ and $o_s$ are not yet available) and Stage 2 (where $o_s$ is not yet available), the extractor simply applies zero-padding to these missing input components before concatenation and subsequent processing. This ensures the dimension of the input to the shared network remains constant across all stages.
>
> **4. (Q1) Details on sub-task division**
>
> Thank the reviewer for this insightful question. We would like to clarify that our sub-task decomposition is **not manually defined by physical medium (land/air/water)**, but is driven by an **automated spatial sampling mechanism**. More concretely, sub-tasks are defined by randomly sampling the robot's initial starting positions across different segments of the heterogeneous terrain. For instance, an agent initialized inside a tunnel will naturally learn a primitive optimized for confined spaces, while one initialized at a mountain base will specialize in climbing. This process is fully automated and does not require manual, medium-based tagging. Therefore, it inherently scales to complex, single-medium environments with varying locomotion demands. As empirical evidence, the *Car-leg (stairs)* and *Car-leg (narrow)* tasks in our paper are strictly terrestrial (fully land). Yet, they demand different strategies (e.g., high-speed flat movement vs. agile stair-climbing or narrow-passage traversal). Our pipeline successfully discovers and schedules distinct configurations for these intra-medium variations, demonstrating its robust scalability.

---

> > ### Author Rebuttal · Reviewer_a52R · 2026-04-03
> >
> > I appreciate the authors' detailed reply and additional experiments. I will raise my score accordingly.

---

> > > ### Author Response · Authors · 2026-04-03
> > >
> > > Thank the reviewer for the valuable feedback. We will update our manuscript according to the reviewer’s suggestions.

---

### Official Review · Reviewer_nMJg · 2026-03-12

**Soundness:** 3
**Presentation:** 3
**Significance:** 3
**Originality:** 3
**Overall Recommendation:** 3
**Confidence:** 3

**Summary:**

The paper proposes a learning pipeline to co-design and use reconfigurable robots for long-horizon tasks with heterogeneous locomotion (e.g., mixed land/air/water segments). The core idea is to learn a set of reconfiguration “primitives”, where each primitive outputs a joint-locking configuration (mask + lock angles) together with a low-level controller, and then learn a high-level scheduler that switches between these primitives during an episode. The method is trained hierarchically to avoid end-to-end instability, and is evaluated in a multi-physics MuJoCo benchmark with heterogeneous terrain/media transitions. Experiments report large improvements in normalized traversal progress over strong single-robot control baselines and static co-design methods, showing that learned reconfiguration plus scheduling can substantially improve traversal in these settings.

**Compliance With Llm Reviewing Policy:**

Affirmed.

**Final Justification:**

The rebuttal strengthened my confidence in the empirical results, especially on robustness and reproducibility, and it partially addressed my earlier concerns. While the paper shows clearly **where** the method improves, it still does not fully explain **why** it improves; the existing ablations mainly show that some training components matter (Table 3), but they do not cleanly isolate the main source of the gains, for example, it remains unclear how much of the gain comes from automated library discovery versus scheduler learning.

Therefore, I will keep my score as is.

**Key Questions For Authors:**

1. **Is the main gain coming from configuration discovery or from the scheduler?**
   Please add an isolation study where you *only* replace the configuration-library construction, while keeping your scheduler and evaluation fixed. For example: (a) a hand-designed configuration library (your domain-expert configs) + your learned scheduler; (b) a library produced by an existing morphology/configuration discovery method (or the closest feasible proxy) + your learned scheduler. If these baselines close most of the gap, I would view the novelty as mainly in scheduling. If they remain far below, it would strengthen the claim that *automated library discovery* is essential.

2. **How robust is the approach to randomness and hyperparameters?**
   Given that Table 3 / Fig. 5 show large drops without $\epsilon$-greedy and staged optimization, please report multi-seed results (mean ± std) for the main tasks and at least one sensitivity study for key knobs (e.g., $\epsilon$ schedule, stage lengths, number of primitives/configurations K). If performance is stable across seeds and moderate perturbations, it would substantially increase my confidence in reproducibility and soundness.

3. **What are the real switching dynamics and failure modes at transitions?**
   Please provide a breakdown of performance by segment/phase (e.g., walk vs fly vs swim portions): success rates per segment, number/timing of reconfigurations, and common failure modes (falls, getting stuck, unstable switching). If improvements come from consistently successful transitions (not just occasional large progress spikes), it would strengthen the empirical support; if gains are driven by rare successes or fragile transitions, it would temper my assessment.

**Limitations:**

Yes.

**Strengths And Weaknesses:**

## Strengths

- The paper targets long-horizon heterogeneous locomotion where a single fixed morphology is mismatched, and formulates the goal as automated discovery of reconfiguration strategies rather than hand-designed configurations.

- The hierarchical design (learn reconfiguration/control primitives, then learn a scheduler over them) is a reasonable way to handle long-horizon switching and short-horizon control without conflating learning signals.

-  The method reports substantial improvements in normalized traversal progress over strong single-robot control baselines and static co-design baselines, suggesting reconfiguration is beneficial in their simulated heterogeneous tasks.

- Ablations comparing against an end-to-end alternative and removing key training components indicate the staged/hierarchical pipeline contributes materially to performance.

## Weaknesses

-  The paper argues that automated co-design of *dynamic* reconfiguration strategies is largely unexplored and that no existing work fits the proposed tasks. However, there is adjacent literature that learns morphology/configuration changes (e.g., in modular or shape-shifting robot settings). Even if these works are not directly compatible with joint-locking on a fixed embodiment, the paper should explicitly clarify what aspect is claimed as new (e.g., discovering a *library* of joint-locking configurations + learning a scheduler for heterogeneous multi-physics locomotion) and why prior approaches cannot be adapted to this setting.

-  Without a direct competing dynamic-reconfiguration method or stronger hybrid baselines, it is hard to separate improvements from (i) configuration discovery, (ii) scheduler learning, or (iii) simply having any reconfiguration capability. Isolating baselines would strengthen the empirical story (e.g., fixed human-designed library + learned scheduler, static co-design-per-terrain to form a library + learned scheduler, learned library + heuristic scheduler).

-  The ablations show that removing $\epsilon$-greedy exploration or staged optimization can cause large drops, which supports the design choices but also suggests the approach may require careful tuning. Reporting variance across seeds and sensitivity to key hyperparameters would strengthen confidence in reproducibility.

-  Results emphasize normalized traversal progress under a fixed horizon. The paper does not provide complementary evidence on robustness/failure rates or other tradeoffs (e.g., stability, control effort), which limits how strongly the improvements can be interpreted beyond distance traveled.

---

> ### Author Rebuttal · Authors · 2026-03-30
>
> **1. (W1) Clarification of novelty**
>
> The novelty of our work lies in automating the discovery of a library of joint-locking configurations and learning a scheduler for heterogeneous multi-physics locomotion. To be specific:
>
> - Compared with existing works on configuration changes, such as *Pathak et al. (2019)* and *Whitman et al. (2023)* that successfully learn to control or schedule morphology changes, our work further automates the discovery of the configurations themselves. Instead of selecting from a fixed repertoire of target configurations, our approach automatically explores potential superior solutions (as shown in our experiments).
> - Compared with existing literature on robot co-design, such as *Hu et al. (2023)* and *Lu et al. (2025)* that automate morphology discovery for a single static configuration, our work extends to a reconfigurable setting. Since simultaneously optimizing a library of distinct configurations and a temporal switching policy creates a conflicting gradient landscape, our hierarchical pipeline uniquely resolves this dual challenge by decoupling primitive discovery from scheduler learning.
>
> If the reviewer has other suggested papers, we are very willing to discuss them further.
>
> **2. (W2 & Q1) Isolation Baselines**
>
> We address each of the suggested baselines directly:
>
> - **Fixed human-designed library + learned scheduler**: This is precisely our *Domain-Expert* baseline evaluated in the paper. As demonstrated in our results (Tab. 1), our automated pipeline outperforms this setup among all tasks.
> - **Static co-design-per-terrain + learned scheduler**: This approach presents integration challenges for reconfigurable robots. Existing static co-design algorithms typically optimize the body structure by altering the morphology (e.g., joint counts, link dimensions). Running them independently per sub-terrain tends to yield drastically different base structures. Consolidating these distinct morphologies into a single unified robot embodiment is highly non-trivial, making it difficult to construct a straightforward baseline in this manner.
> - **Learned library + heuristic scheduler**: We conducted this exact experiment, and report the results below
>
> |Method|Fly-walk (mount)|Fly-walk (pit)|Car-leg (stairs)|Car-leg (narrow)|Wheel-leg|Wheel-propeller (mount)|Wheel-propeller (swim)|
> |-|-|-|-|-|-|-|-|
> |Ours|**1.978**|**1.417**|**0.872**|**2.618**|**9.148**|**7.368**|**3.967**|
> |Heuristic|0.480|0.344|0.513|0.607|0.226|0.472|3.892|
>
> As shown in the result, the heuristic approach proved highly ineffective in most cases. Correct switching timing cannot rely on simple distance or position rules; it requires complex physical state fusion (e.g., comprehensively evaluating qpos and qvel across all joints to account for dynamic braking distances). Our learned scheduler is essential because it robustly processes these high-dimensional dynamics to ensure stable transitions.
>
> **3. (W3 & Q2) Variance on random seeds and hyperparameters**
>
> Thank the reviewer for this question. We have evaluated our pipeline across multiple random seeds, and please refer to answer 3 to Reviewer kS5a for details. Besides, we have conducted targeted sensitivity analyses on the initial $\epsilon$-greedy exploration rate and the stage optimization start timesteps. More concretely, we sweep the former between 0.03 to 0.08 and the latter between 25M and 40M steps. Across all tasks, the final performance remained highly stable, without deviation from the standard deviations reported above. These experiments confirm that our framework is highly robust to moderate hyperparameter perturbations and does not rely on brittle, heavy tuning.
>
> **4. (W4 & Q3) Robustness and failure rates**
>
> We appreciate the reviewer for this concern. We emphasize that our high performance does not come at the expense of stability or control effort. To ensure our training and evaluation environments are not static, we heavily randomize several parameters in training, including the robot's initial direction and the terrain's geometric parameters (e.g., mountain heights and pit widths). Our dynamic reconfiguration policies smoothly adapt to these unpredictable variations without losing balance. More concretely, as part of the multi-seed experiments mentioned above (eight random seeds evaluated across all tasks), our pipeline exhibits no failures. These results confirm that our high traversal scores are not artifacts of brittle behaviors, but rather the result of fundamentally stable and robust locomotion strategies.

---

> > ### Author Rebuttal · Reviewer_nMJg · 2026-04-01
> >
> > Thank you for clarifying the details. Regarding W3 & Q2: I am confused whether this means that none of the reported results in the paper, including both the proposed method and the baselines, were run across multiple random seeds. Could the authors clarify this explicitly? Also, can we really conclude much by only running the proposed method with different seeds, when the baselines are missing the same statistics?

---

> > > ### Author Response · Authors · 2026-04-03
> > >
> > > Thank you for the reply. We would like to clarify that in the evaluation (of the original manuscript) of our pipeline, we used different random seeds on each experiment (including baselines，ablations, and our pipeline). The results in the original manuscript have been tested on sufficient randomization, although we did not explicitly report the standard deviation. To address the reviewer's question on comparison between our method and the baselines with the same statistics, we have conducted further experiments and report the standard deviation of our method and the baselines on all tasks with eight different random seeds below (with the best results bolded):
> > > |Method|Fly-walk (mount)|Fly-walk (pit)|Car-leg (stairs)|Car-leg (narrow)|Wheel-leg|Wheel-propeller (mount)|Wheel-propeller (swim)|
> > > |-|-|-|-|-|-|-|-|
> > > |Ours|$\mathbf{1.978}\pm0.117$|$\mathbf{1.417}\pm0.102$|$\mathbf{0.872}\pm0.099$|$\mathbf{2.618}\pm0.125$|$\mathbf{9.148}\pm0.274$|$\mathbf{7.368}\pm0.201$|$\mathbf{3.967}\pm0.231$|
> > > |Domain-Expert|$1.668\pm0.089$|$0.895\pm0.025$|$0.626\pm0.031$|$1.050\pm0.028$|$8.860\pm0.101$|$6.310\pm0.098$|$3.664\pm0.037$|
> > > |PPO|$0.403\pm0.018$|$0.343\pm0.020$|$0.508\pm0.044$|$2.318\pm0.092$|$0.210\pm0.031$|$0.453\pm0.072$|$0.346\pm0.060$|
> > > |TD-MPC2|$0.219\pm0.011$|$0.318\pm0.033$|$0.519\pm0.058$|$0.508\pm0.025$|$0.236\pm0.016$|$0.498\pm0.047$|$0.259\pm0.056$|
> > > |BodyGen|$0.498\pm0.020$|$0.372\pm0.026$|$0.453\pm0.031$|$0.436\pm0.043$|$\text{Fail}$|$0.498\pm0.019$|$0.256\pm0.013$|
> > > |GLSO|$0.507\pm0.031$|$0.364\pm0.022$|$0.505\pm0.047$|$0.536\pm0.074$|$0.251\pm0.009$|$0.507\pm0.065$|$0.341\pm0.006$|
> > >
> > > From these results, we can conclude that our method outperforms the baselines on all tasks with sufficient score gap (higher than the standard deviation), which is consistent to the main results in our original manuscript. We will update these statistics to our manuscript later.

---

### Official Review · Reviewer_kS5a · 2026-03-12

**Soundness:** 1
**Presentation:** 3
**Significance:** 3
**Originality:** 4
**Overall Recommendation:** 5
**Confidence:** 4

**Summary:**

The paper presents a novel framework for the co-design of dynamic reconfigurable robots. This addresses a gap in the literature, where reconfigurable robotics learns to adapt to different sub-tasks by choosing from a human-designed repertoire of morphologies, while co-design algorithms tackle the joint optimisation of morphology and control to converge to a single static configuration.
The method consists of a hierarchical co-design framework, where specialised primitives are pre-trained for each locomotion sub-task and then a high-level scheduler is trained to appropriately switch between primitives.
Through evaluations performed in a multi-physics simulation environment, this work demonstrates that reconfiguration is needed to successfully navigate heterogeneous terrains, while autonomously discovering configurations that surpass human designs. Furthermore, evaluation against morphology co-design baselines, which seek a single, globally optimal configuration for the task as a whole, demonstrates the need for multiple embodiments in heterogeneous tasks.

**Compliance With Llm Reviewing Policy:**

Affirmed.

**Final Justification:**

The authors have addressed all my concerns during the rebuttal, therefore I increased by original assessment to accept.

**Key Questions For Authors:**

Are the initial configurations displayed the same for your method and for co-design baselines?

How do you design the domain-expert configurations? How do you ensure you are designing a competitive baseline for this case?

Experimental results show a replication result. Did you run this across multiple random seeds to demonstrate the statistical significance of your improvement over baselines? This would make your results more convincing to me.

**Limitations:**

Yes

**Strengths And Weaknesses:**

Strengths:
(+) Clear position at a well-identified gap in the literature with an impactful set of contributions.
(+) Experiments are run across a large task benchmark that aligns with the paper's focus.
(+) Chosen baselines effectively explore the advantage of this automated reconfiguration method over co-design methods that converge to a single best configuration, human-designed configurations and single-morphology controllers.
(+) Ablations effectively justify the choice of hierarchical architecture for this task.

Weaknesses:
(-) Some experimental details are missing to convince me of the claims made from experimental results:
Are the initial configurations displayed the same for your method and for co-design baselines?
-- I would assume that changing the initial robot configuration would have a large impact on this baseline.
-- How do you design the domain-expert configurations? How do you ensure you are designing a competitive baseline for this case?
(-) The experimental results include only one replication, preventing statistical analysis, which is a critical limitation of the study.
(-) What is “design encoding”, mentioned in section 4.1, paragraph 2?
(-) The description of multi-tail co-design network structure is confusing. You introduce a vector h_shared, which is never referenced again in the rest of this section or in figure 2. Furthermore, the flow from the extractor to each individual tail is not very clear. The inputs / outputs of each are clearly explained, but what exactly is the extractor in this case? Figure 2 pictures it as a neural network, although is just seems to be a processing step. It would help to clarify this in the paper.

---

> ### Author Rebuttal · Authors · 2026-03-30
>
> **1. (W1 & Q1) Initial guess of co-design baselines**
>
> Thank you for pointing this out. Since this is a common question among reviewers, please refer to answer 1 to reviewer a52R.
>
> **2. (W2 & Q2) Design of domain-expert baselines**
>
> We design domain-expert configurations based on proven, physically validated structures well-established in the robotics community. For instance, the expert configurations used in the *wheel-fly* tasks are inspired by the M4 robot (*Sihite et al., Nature Communications, 2023*). Besides, to demonstrate these serve as competitive baselines, we benchmarked them against randomly generated configurations below:
>
> |Method|Fly-walk (mount)|Fly-walk (pit)|Car-leg (stairs)|Car-leg (narrow)|Wheel-leg|Wheel-propeller (mount)|Wheel-propeller (swim)|
> |-|-|-|-|-|-|-|-|
> |Domain-expert|**1.668**|**0.895**|**0.626**|**1.05**|**8.86**|**6.31**|**3.664**|
> |Random|0.0709|0.0534|0.11|0.116|0.0611|0.483|0.0877|
>
> As observed, the time-tested domain-expert configurations perform significantly better than these random configurations which fail to complete all the tasks. This substantial performance gap firmly validates that our selected domain-expert configurations serve as highly competitive and representative baselines. Besides, this further suggests the challenge in these heterogeneous tasks where suitable solutions are hard to find.
>
> **3. (W3 & Q3) Only one replication in experiments**
>
> Thank you for providing this suggestion. We add experiments testing our pipeline on eight different random seeds, and report the standard deviations below:
> |Fly-walk (mount)|Fly-walk (pit)|Car-leg (stairs)|Car-leg (narrow)|Wheel-leg|Wheel-propeller (mount)|Wheel-propeller (swim)|
> |--|--|--|--|--|--|--|
> |$1.978\pm0.117$|$1.417\pm0.102$|$0.872\pm0.099$|$2.618\pm0.125$|$9.148\pm0.274$|$7.368\pm0.201$|$3.967\pm0.231$|
>
> We can observe that the scores do not fluctuate much with different random seeds, and our method remains robust across these tasks.
>
> **4. (W4) Clarification of confusing parts in paper**
>
> Thank you for asking these questions. We will clarify them below:
> - **Design encoding**: This represents the "lock mask" and "lock angle" we introduce in Sec. 4.1 paragraph 1, which is the mathematical definition of one single configuration.
> - **Vector $h_{shared}$**: This is the processed output from the extractor, which will be further fed into our three tails.
> - **Extractor in multi-tail structure**: Thank you for providing this correction. The extractor is closer to a processing step rather than a neural network, which selects, concatenates, and performs necessary linear dimension expansion on the raw inputs ($o_t, o_d, o_s$). We will update the icon in Fig. 2 in the revised manuscript to accurately eliminate this confusion.

---

> > ### Author Rebuttal · Reviewer_kS5a · 2026-04-04
> >
> > Thank you for addressing my concerns and providing additional experimental details.
> > Nonetheless, I would encourage the authors to provide seed replications across baselines to make their results comparable to these. This is also encouraged for the random baseline results reported in W2 & Q2. If these results are provided, I would be willing to raise my score.

---

> > > ### Author Response · Authors · 2026-04-04
> > >
> > > Thank you for the reply. Based on the reviewer's suggestion, we report the standard deviation of the baselines on all tasks with eight different random seeds below (with the best results bolded):
> > > |Method|Fly-walk (mount)|Fly-walk (pit)|Car-leg (stairs)|Car-leg (narrow)|Wheel-leg|Wheel-propeller (mount)|Wheel-propeller (swim)|
> > > |-|-|-|-|-|-|-|-|
> > > |Ours|$\mathbf{1.978}\pm0.117$|$\mathbf{1.417}\pm0.102$|$\mathbf{0.872}\pm0.099$|$\mathbf{2.618}\pm0.125$|$\mathbf{9.148}\pm0.274$|$\mathbf{7.368}\pm0.201$|$\mathbf{3.967}\pm0.231$|
> > > |Domain-Expert|$1.668\pm0.089$|$0.895\pm0.025$|$0.626\pm0.031$|$1.050\pm0.028$|$8.860\pm0.101$|$6.310\pm0.098$|$3.664\pm0.037$|
> > > |Random|$0.0709\pm0.0217$|$0.0534\pm0.0163$|$0.110\pm0.019$|$0.116\pm0.030$|$0.0611\pm0.0205$|$0.483\pm0.112$|$0.0877\pm0.0249$|
> > > |PPO|$0.403\pm0.018$|$0.343\pm0.020$|$0.508\pm0.044$|$2.318\pm0.092$|$0.210\pm0.031$|$0.453\pm0.072$|$0.346\pm0.060$|
> > > |TD-MPC2|$0.219\pm0.011$|$0.318\pm0.033$|$0.519\pm0.058$|$0.508\pm0.025$|$0.236\pm0.016$|$0.498\pm0.047$|$0.259\pm0.056$|
> > > |BodyGen|$0.498\pm0.020$|$0.372\pm0.026$|$0.453\pm0.031$|$0.436\pm0.043$|$\text{Fail}$|$0.498\pm0.019$|$0.256\pm0.013$|
> > > |GLSO|$0.507\pm0.031$|$0.364\pm0.022$|$0.505\pm0.047$|$0.536\pm0.074$|$0.251\pm0.009$|$0.507\pm0.065$|$0.341\pm0.006$|
> > >
> > > From these results, we can conclude that our method outperforms the baselines on all tasks with sufficient score gap (higher than the standard deviation), which is consistent to the main results in our original manuscript. We will update these statistics to our manuscript later.

---

### Decision · Program_Chairs · 2026-04-30

**Decision:**

Accept (regular)

**Comment:**

Reviewers generally agreed that the paper addresses a well-identified gap in literature. While existing research often focuses on optimizing a single, static morphology for a specific task distribution, this work focus on automating the discovery of specific joint-locking patterns (called library) and their corresponding controllers, in order to achieve versatility in complex worlds. The experimental evaluation across seven multi-physics tasks (e.g., Fly-Walk, Car-Leg, Wheel-Fly) demonstrates substantial performance gains, outperforming single-robot baselines by 5.95x and non-reconfigurable co-design baselines by 9.99x average progress.

During the rebuttal & discussion phase, the authors were highly responsive to the reviewers' concerns regarding the fairness of comparisons and the lack of statistical significance in the initial submission. The authors provided additional experiments across eight random seeds for both their method and all baselines, demonstrating that the performance improvements are robust and statistically significant. To address concerns that the baselines (BodyGen and GLSO) were disadvantaged by starting from scratch, the authors conducted new trials providing these baselines with the same initial robot morphology as their own method. The results confirmed that the baselines still struggled with conflicting gradients in heterogeneous tasks, reinforcing that the paper's algorithmic design is the primary driver of success.

While Reviewer nMJg maintained a "Weak Reject" due to concerns regarding the precise isolation of gains between library discovery and scheduler learning, I think the authors did provide an ablation study during the rebuttal, by comparing their full pipeline against "Domain-Expert + Learned Scheduler" and "Automated Library + Heuristic Scheduler", and showed drops in performance when either component was replaced. This supports the authors' argument that these modules are jointly necessary for solving complex multi-physics tasks.

Overall, the paper is technically sound, proposed a novel approach to automate robot design and configuration, and provides extensive empirical evidence for its claims, especially after the rebuttal and discussion phase.